# Learning to Defer with an Uncertain Rejector via Conformal Prediction

**Yizirui Fang**                                                                    *yfang52@jhu.edu*
*Department of Computer Science*
*Johns Hopkins University*

**Eric Nalisnick**                                                                  *nalisnick@jhu.edu*
*Department of Computer Science*
*Johns Hopkins University*

**Reviewed on OpenReview:** *https://openreview.net/forum?id=SZQJ8K2DUe*

## Abstract

*Learning to defer* (L2D) aims to optimize human-AI collaboration by allocating prediction tasks to either a machine learning model or a human expert, depending on which is most likely to be correct. This allocation decision is governed by a *rejector*: a meta-model that routes inputs based on estimated success probabilities. In practice, a poorly fit or otherwise misspecified rejector can jeopardize the entire L2D workflow due to its crucial role in allocating prediction tasks. In this work, we perform uncertainty quantification for the rejector. We use conformal prediction to allow the rejector to output prediction sets or intervals instead of just the binary outcome of 'defer' or not. On tasks ranging from image to hate speech classification, we demonstrate that the uncertainty in the rejector translates to safer decisions via two forms of selective prediction.

## 1 Introduction

*Learning-to-Defer* (L2D) (Madras et al., 2018; Mozannar & Sontag, 2020) is a framework for human-AI collaboration that divides responsibility between machine and human decision makers. For every test instance, a 'rejector' function decides if the case should be passed to either a human or model (but not both). The rejector can be seen as a meta-classifier that determines how to assign responsibility based on which decision maker (human or machine) is more likely to make the correct prediction. L2D systems offer the promise of improved safety and robustness by having a human available for support. Yet this promise critically depends on the rejector's performance. Being a predictive model itself, the rejector is susceptible to the usual failure modes, such as distribution shift between training and test data and label noise. Yet, unlike with traditional predictive models, there is an extra point of failure in that the distribution of the human's predictions can also shift (Tailor et al., 2024).

In this paper, we perform principled uncertainty quantification (UQ) for the rejector sub-component of L2D systems. Specifically, we use the framework of *conformal prediction* (CP) (Vovk et al., 2022) to allow the rejector to output sets or intervals, instead of just a binary outcome (defer or not). This allows the rejector to express its uncertainty about whether the human or machine should be assigned to make the decision. In turn, this unlocks new abilities for the L2D system. For example, if the rejector is unsure about to whom responsibility should be assigned, the system can simply abstain from making any prediction. As a concrete example, consider automated diagnosis of skin lesions. For each dermoscopic image, our L2D system either outputs a prediction or abstains, depending on if the allocation decision was sufficiently certain. In the latter case, a clinician could request further information such as running additional lab tests or biopsies before making the final decision. Alternatively, an uncertain rejector could mean that we should query *both* the human and model for their predictions. If the human and model agree on their prediction, then this is a good indication that that prediction is reliable (à la ensembling).

We explore these novel L2D workflows in experiments on tasks ranging from image to hate speech classification. We find that introducing UQ into L2D allows safe alternative behaviors (e.g. abstention or consensus checking, as mentioned above) that prevent the L2D system from otherwise returning the wrong prediction. We also study various L2D learning objectives, parameterizations, and CP formulations, finding that the one-vs-all parameterization tends to result in better downstream performance (e.g. accuracy) but at the cost of sometimes having too small of uncertainty sets and in turn under-covering the true label. In summary, we make the following contributions:[1]

- **Distribution-Free UQ for L2D Allocation** (Sections 3 - 3.1): We are the first to formulate a UQ problem for the L2D deferral decision. Moreover, we are the first to apply conformal prediction to the rejector sub-model of L2D, providing distribution-free coverage guarantees on expert correctness.

- **Novel L2D Workflows** (Section 3.2 - 3.3): We propose four novel, alternative workflows for L2D that operate via (i) abstention, (ii) checking for consensus between expert and model predictions, (iii) preferring to query the model when the human is uncertain (for cost saving), and (iv) preferring to query the human under distribution shift.

## 2 Background

We first review the necessary background information on L2D and conformal prediction.

### 2.1 Learning to Defer

**Setting, Data, and Model**  We focus on multiclass L2D (with one expert) (Madras et al., 2018; Mozannar & Sontag, 2020), though the ideas presented can straightforwardly generalize to L2D-based regression (Zaoui et al., 2020). Let $\mathcal{X}$ denote the feature space and $\mathcal{Y}$ the label space, a categorical encoding of $K \in \mathbb{N}^{\geq 2}$ classes. Let $\mathbf{x}_n \in \mathcal{X}$ denote a feature vector, and $\mathrm{y}_n \in \mathcal{Y}$ denotes the associated class index. L2D assumes that we have access to human predictions, denoted $\mathrm{m}_n \in \mathcal{Y}$ for the associated feature vector $\mathbf{x}_n$. The training data then includes the features, the true label, and the human's prediction: $\mathcal{D} = \{\boldsymbol{x}_n, y_n, m_n\}_{n=1}^N$. The human is assumed to be skilled at the prediction task but is not an oracle. For example, the feature vector could be a medical image, $m_n$ is the expert's diagnosis from looking at the image, and $y_n$ is a true label that can only be obtained from a biopsy. L2D also assumes that the human has access to background knowledge that the classifier does not, such as years of medical training in the aforementioned example. The L2D framework requires two sub-models: a classifier and a rejector (Cortes et al., 2016b;a). We denote the *classifier* as $h : \mathcal{X} \to \mathcal{Y}$ and the *rejector* as $r : \mathcal{X} \to \{0, 1\}$. When $r(\mathbf{x}) = 0$, the classifier makes the decision, and when $r(\mathbf{x}) = 1$, the classifier abstains and defers the decision to the human. Thus the rejector can be thought of as a 'meta-classifier,' predicting which *predictor* would most likely be correct in its prediction.

**Learning**  Learning in L2D requires us to fit both the rejector and classifier. We assume that whoever makes the prediction—model or human—incurs a loss of zero (correct) or one (incorrect). Using the rejector to toggle between the human and model, we have the overall classifier-rejector loss:

$$L_{0-1}(h, r) = \mathbb{E}_{\mathbf{x},\mathrm{y},\mathrm{m}}\Big[(1 - r(\mathbf{x})) \cdot \mathbb{I}\left[h(\mathbf{x}) \neq \mathrm{y}\right] + r(\mathbf{x}) \cdot \mathbb{I}[\mathrm{m} \neq \mathrm{y}]\Big] \tag{1}$$

where $\mathbb{I}\left[h(\mathbf{x}) \neq \mathrm{y}\right]$ denotes an indicator function that the model prediction does not match the label and $\mathbb{I}[m \neq \mathrm{y}]$ denotes another indicator function that the human prediction does not match the label. Minimizing this loss results in the Bayes optimal classifier and rejector:

$$h^*(\boldsymbol{x}) = \underset{y \in \mathcal{Y}}{\arg\max} \ \mathbb{P}(\mathrm{y} = y|\boldsymbol{x}), \qquad r^*(\boldsymbol{x}) = \mathbb{I}\left[\mathbb{P}(\mathrm{m} = \mathrm{y}|\boldsymbol{x}) > \underset{y \in \mathcal{Y}}{\max} \mathbb{P}(\mathrm{y} = y|\boldsymbol{x})\right] \tag{2}$$

where $\mathbb{P}(\mathrm{y}|\boldsymbol{x})$ is the probability of the label under the data generating process and $\mathbb{P}(\mathrm{m} = \mathrm{y}|\boldsymbol{x})$ is the probability that the expert is correct. The assumption that the expert has additional knowledge is what allows them to possibly outperform the Bayes optimal classifier.

---

[1] An abbreviated version of this paper appeared in the non-archival proceedings of the NeurIPS 2024 Workshop on *Bayesian Decision-Making and Uncertainty*.

**Surrogate Losses**  Several consistent surrogate losses have been proposed for Equation 1 (Mozannar & Sontag, 2020; Verma & Nalisnick, 2022; Mao et al., 2024c;b; Cao et al., 2023; Charusaie et al., 2022). For our implementation, we focus on the two surrogates that have demonstrated the ability to learn calibrated predictors in practice since the more calibrated the predictor, the better the conformal prediction results will be. Specifically, we use Verma & Nalisnick (2022)'s one-vs-all (OvA) parameterization and Cao et al. (2023)'s asymmetric softmax (A-SM) parameterization. These parameterizations assume the classifier and rejector are unified via an augmented label space: $\mathcal{Y}^{\perp} = \mathcal{Y} \cup \{\perp\}$, where $\perp$ denotes the rejection option. Then let $g_k : \mathcal{X} \mapsto \mathbb{R}$ for $k \in [1, K]$ where $k$ denotes the class index, and let $g_{K+1} : \mathcal{X} \mapsto \mathbb{R}$ denote the rejection ($\perp$) option. The $g$ functions are analogous to the logits of a neural-network-based classifier. The OvA surrogate loss is given as (Verma & Nalisnick, 2022):

$$\psi_{\text{OvA}}(g_1, \ldots, g_{K+1}; \boldsymbol{x}, y, m) = \phi[g_y(\boldsymbol{x})] + \sum_{y' \in \mathcal{Y}, y' \neq y} \phi[-g_{y'}(\boldsymbol{x})] + \phi[-g_{K+1}(\boldsymbol{x})] + \mathbb{I}[m = y] \cdot (\phi[g_{K+1}(\boldsymbol{x})] - \phi[-g_{K+1}(\boldsymbol{x})]) \tag{3}$$

where $\phi : \{\pm 1\} \times \mathbb{R} \mapsto \mathbb{R}_+$ is a binary surrogate loss. For instance, when $\phi$ is the logistic loss, we have $\phi[f(\boldsymbol{x})] = \log(1 + \exp\{-f(\boldsymbol{x})\})$.

The A-SM surrogate loss is defined as follows (Cao et al., 2023):

$$\psi_{\text{A-SM}}(g_1, \ldots, g_{K+1}; \boldsymbol{x}, y, m) = -\log \phi_{\text{A-SM}}(g(\boldsymbol{x}), y) - \mathbb{I}[m \neq y] \cdot \log(1 - \phi_{\text{A-SM}}(g(\boldsymbol{x}), K+1)) - \mathbb{I}[m = y] \cdot \log \phi_{\text{A-SM}}(g(\boldsymbol{x}), K+1) \tag{4}$$

$$\text{where} \quad \phi_{\text{A-SM}}(g(\boldsymbol{x}), y) = \begin{cases} \dfrac{\exp\{g_y(\boldsymbol{x})\}}{\sum_{y'=1}^{K} \exp\{g_{y'}(\boldsymbol{x})\}} & \text{if } y \neq K+1, \\[2ex] \dfrac{\exp\{g_{K+1}(\boldsymbol{x})\}}{\sum_{y'=1}^{K+1} \exp\{g_{y'}(\boldsymbol{x})\} - \max_{y' \in \mathcal{Y}} \exp\{g_{y'}(\boldsymbol{x})\}} & \text{otherwise.} \end{cases}$$

Here, the 'asymmetry' is due to the softmax having different terms in the denominator for the class and rejector terms. The symmetric softmax parameterization (Mozannar & Sontag, 2020) has the same denominator for both terms, which leads to issues for estimating the expert's correctness probability in practice (Verma & Nalisnick, 2022; Cao et al., 2023). For both parameterizations, at test time, the classifier is obtained by taking the maximum over $g$ functions: $\hat{y} = h(\boldsymbol{x}) = \arg\max_{y' \in [1, K]} g_{y'}(\boldsymbol{x})$. The rejector is implemented as: $r(\boldsymbol{x}) = \mathbb{I}[g_{K+1}(\boldsymbol{x}) \geq \max_{y' \in \mathcal{Y}} g_{y'}(\boldsymbol{x})]$.

**Expert Correctness**  Both the OvA and A-SM parameterizations compute the probability that the expert is correct. For the OvA parameterization, this probability is directly parameterized by the $(K+1)$th binary classifier:

$$\hat{p}(\text{m} = \text{y}|\mathbf{x}) = \phi_{\text{OvA}}[g_{K+1}(\mathbf{x})] = \frac{1}{1 + \exp\{-g_{K+1}(\boldsymbol{x})\}}. \tag{5}$$

A-SM similarly uses the deferral score, but here the parameterization requires evaluating all $K+1$ functions:

$$\hat{p}(\text{m} = \text{y}|\mathbf{x}) = \phi_{\text{A-SM}}(g(\boldsymbol{x}), K+1) = \frac{\exp\{g_{K+1}(\boldsymbol{x})\}}{\sum_{y'=1}^{K+1} \exp\{g_{y'}(\boldsymbol{x})\} - \max_{y' \in \mathcal{Y}} \exp\{g_{y'}(\boldsymbol{x})\}}. \tag{6}$$

Both estimators have been shown to be competitively calibrated when trained by empirical risk minimization and without relying upon post-hoc procedures such as temperature scaling (though they could be employed as well) (Cao et al., 2023).

## 2.2  Conformal Prediction

*Conformal prediction* (CP) is a model-agnostic, distribution-free approach to uncertainty quantification with finite-sample guarantees (Shafer & Vovk, 2008; Angelopoulos & Bates, 2023). Given a test-time feature vector $\mathbf{x}_{N+1}$, CP seeks to construct a prediction set $C(\mathbf{x}_{N+1}; \tau) \subseteq \mathcal{Y}$ such that the true label $\text{y}_{N+1}$ is included with

probability $1 - \alpha$: $\mathbb{P}\left(y_{N+1} \in C\left(\mathbf{x}_{N+1}; \tau\right)\right) \geq 1 - \alpha$, for $\alpha \in [0, 1]$. $\tau$ is a parameter that controls the set size, as will be described below. This statement is a *marginal* guarantee, meaning that it will hold, on average, over test samples but will not necessarily hold for any particular sample. CP's aforementioned guarantee is built off the crucial assumption that the test data is drawn exchangeably with a calibration set.

To compute the parameter $\tau$ that controls the prediction sets, the *split*-CP (a.k.a. *inductive* CP) algorithm (Papadopoulos et al., 2002) is a popular choice due to its computational and sample efficiency (Fang & Bellotti, 2024) and resemblance to the traditional workflow of hyperparameter tuning. Split-CP requires $\tau$ be fit to a held-out calibration set $\mathcal{D}_2$, which must be drawn exchangeably with the test set for the CP coverage guarantee to hold. Given a classifier already trained on the training set $\mathcal{D}_1$, its estimated class probabilities are denoted $\boldsymbol{f}(\mathbf{x}) = [f_1(\mathbf{x}), \ldots, f_K(\mathbf{x})]$. CP then requires a score function be chosen that quantifies how well the model's prediction conforms to the true label's. Using the softmax confidence associated with the true label is a reasonable choice: $s\left(\mathbf{x}, \mathbf{y}; \boldsymbol{f}\right) = 1 - f_{\mathbf{y}}(\mathbf{x})$, where $f_{\mathbf{y}}(\mathbf{x})$ is the estimated score for the true label. Others exist that incorporate all dimensions that have higher confidence than the true label (Romano et al., 2020). Split-CP then proceeds by evaluating $s\left(\mathbf{x}, \mathbf{y}; \boldsymbol{f}\right)$ on all points in the held-out set and setting $\hat{\tau}$ to be the $(1 - \alpha)$ quantile (with a finite-sample correction) of the empirical distribution of scores. For a test time point $\mathbf{x}_{N+1}$, the prediction set is constructed as:

$$C(\mathbf{x}_{N+1}) \;=\; \{\; j \;\mid\; f_j(\mathbf{x}_{N+1}) \geq 1 - \hat{\tau} \;\},$$

which represents the softmax dimensions that outscore the threshold $1 - \hat{\tau}$. CP is commonly evaluated by checking that the desired coverage level is achieved in practice while also having efficient set sizes. The latter is crucial since the CP guarantee is trivially met by choosing $C(\mathbf{x}_{N+1}; \tau) = \mathcal{Y}$ for $(1 - \alpha)\%$ of test cases.

## 3   Uncertain Deferral via Conformal Prediction

We will now apply the CP framework to quantify the uncertainty in the rejector sub-component of an L2D system. Concretely, instead of just outputting 0 (model) or 1 (human), we want the CP-based rejector to output a set $C_r\left(\mathbf{x}; \tau\right)$, which is an element of the superset $\{\{0\}, \{1\}, \{0, 1\}\}$. $C_r\left(\mathbf{x}; \tau\right) = \{0, 1\}$ means that the rejector is unsure if the decision should be allocated to the human or model. Thus, instead of *prediction* sets, we call the uncertainty set of the rejector a *deferral set*. In Section 3.2, we will discuss how to incorporate these sets into downstream decision making.

**Ideal Construction**   Recalling the Bayes optimal decision rule for the rejector (Equation 2), it would be ideal if $C_r\left(\mathbf{x}; \tau\right)$ could satisfy the guarantee:

$$\mathbb{P}\left(r^*\left(\mathbf{x}_{N+1}\right) \;\in\; C_r\left(\mathbf{x}_{N+1}; \tau\right)\right) \;\geq\; 1 - \alpha,$$

which means that, marginally, the probability that the output of the Bayes optimal rejector is in the set is at least $1 - \alpha$. Constructing an adaptive set via validation statistics, unfortunately, requires that we be able to compare $\mathbb{P}(\mathbf{m} = \mathbf{y}|\boldsymbol{x})$ vs $\mathbb{P}(\mathbf{y}|\boldsymbol{x})$ to compute a non-conformity score. This comparison requires high-fidelity estimates of two conditional probabilities, and obtaining estimates of such one-off events is known to be impossible (Roth et al., 2023). The only work-around is if we observe multiple samples of both the label y and expert prediction m (Johnson et al., 2024), which would only drastically increase the already high supervision burden of L2D. Thus, we leave this construction as an open problem and turn to a more practical alternative below.

**Practical Construction**   We instead consider constructing the set to capture an alternative quantity: $\mathbb{I}\left[\mathbf{m}_{N+1} = \mathbf{y}_{N+1}\right]$, an indicator function representing if the human will make the correct prediction. Similarly, we wish to construct prediction sets such that this binary variable will have a coverage guarantee:

$$\mathbb{P}\left(\mathbb{I}\left[\mathbf{m}_{N+1} = \mathbf{y}_{N+1}\right] \;\in\; C_r\left(\mathbf{x}_{N+1}; \tau\right)\right) \geq 1 - \alpha. \tag{7}$$

This statement is not equivalent to the one above since the expert could be correct (i.e. $\mathbb{I}\left[\mathbf{m}_{N+1} = \mathbf{y}_{N+1}\right] = 1$) but $\mathbb{P}(\mathbf{y}|\mathbf{x})$ still be a better predictive model (i.e. $r^*(\mathbf{x}) = 0$). In other words, this formulation is considering

the expert's performance in isolation of the classifier's. However, the high-level semantics are retained since $C_r(\mathbf{x}_{N+1}; \tau) = \{0\}$ means that the expert will likely be wrong and so using the classifier is either a good decision or not an inferior one (if the model would also be wrong). Conversely, $C_r(\mathbf{x}_{N+1}; \tau) = \{1\}$ means that the expert will likely make the correction prediction. If $C_r(\mathbf{x}_{N+1}; \tau) = \{0, 1\}$, then the prediction set is unsure if the expert will be correct and still suggests uncertainty in the deferral decision. This relaxation, importantly, allows us to define a practical conformity statistic. Prior work has demonstrated that this practical construction—estimating expert correctness—can be properly calibrated and effectively correlates with the expert's superiority over the classifier (Verma & Nalisnick, 2022).

## 3.1 Constructing Deferral Sets

We can construct deferral sets that follow the guarantee in Equation 7 by treating the deferral decision as a binary classification problem: whether the expert will make the correct prediction or not. Following CP as it is usually applied to binary classification, we construct the conformity score using the binary probabilities given in Equation 5 for OvA and Equation 6 for A-SM. To obtain the threshold $\hat{\tau}$, we follow the standard procedure of split-CP by computing these non-conformity scores on a held-out calibration set (Angelopoulos & Bates, 2023), obtaining the $(1 - \alpha)$ empirical quantile, and applying the threshold at test time as follows:

$$C_r(\mathbf{x}; \hat{\tau}) = \begin{cases} \{0\} & \text{if } 1 - \hat{p}(\mathrm{m} = \mathrm{y}|\mathbf{x}) \geq 1 - \hat{\tau} \\ \{1\} & \text{if } \hat{p}(\mathrm{m} = \mathrm{y}|\mathbf{x}) \geq 1 - \hat{\tau} \\ \{0, 1\} & \text{otherwise.} \end{cases} \tag{8}$$

The set $C_r(\mathbf{x}; \hat{\tau})$ should satisfy the coverage guarantee given in Equation 7, assuming the usual assumptions of CP hold, such as exchangeability between calibration and test data.

## 3.2 Using Deferral Sets in Decision Making

Now that we have detailed how to construct CP deferral sets, we next address how to use them to improve decision making within the L2D framework. While there are surely alternative uses, below we detail three that we believe will be practical and useful in a variety of applications.

**Abstention** The use that likely first comes to mind is prediction with the option to abstain (Chow, 1957; Cordella et al., 1995; Herbei & Wegkamp, 2006; Hellman, 1970; Geifman & El-Yaniv, 2017). In the traditional case, the classifier only makes a prediction if it is confident; otherwise, it abstains since the consequences of being wrong outweigh the consequences of providing no prediction. This is often appropriate for applications in healthcare: it is better to wait and perform more tests, seek out more opinions, etc. than to give a patient a wrong diagnosis. Our CP deferral sets allow for a similar workflow, but instead of abstaining because the prediction is uncertain, the L2D system will abstain because it is uncertain about to whom to allocate responsibility, the machine or human. Specifically, if $C_r(\boldsymbol{x}_{N+1}; \hat{\tau}) = \{0, 1\}$, then the L2D system will abstain. Otherwise, the system will defer if $r^*(\boldsymbol{x}) = 1$. A visualization of this workflow is shown in Figure 1a. As is typically the case with abstention methods, we expect this workflow to improve the system accuracy at the cost of reducing coverage.

**Consensus Prediction** We next consider how to make a prediction even if $C_r(\boldsymbol{x}_{N+1}; \hat{\tau}) = \{0, 1\}$. If the rejector is uncertain to defer or not, we propose querying both the model and human for their predictions. If they agree, then that consensus prediction is output as the L2D system's final prediction. If they do not agree, then the system abstains from making any prediction. This workflow has the same appeal to safety as the abstention-only option, but it will have higher coverage since it will make predictions when the abstention-only workflow would not. This workflow is diagrammed in Figure 1b. We expect this workflow to perform similarly as abstention but with increased coverage, since it can still make predictions even when the deferral set is of maximum size.

**Human-Preferred Prediction under Distribution Shift** We consider cases in which intuition leads us to believe the human is a more robust predictor than the model. Consider the task of image classifier under

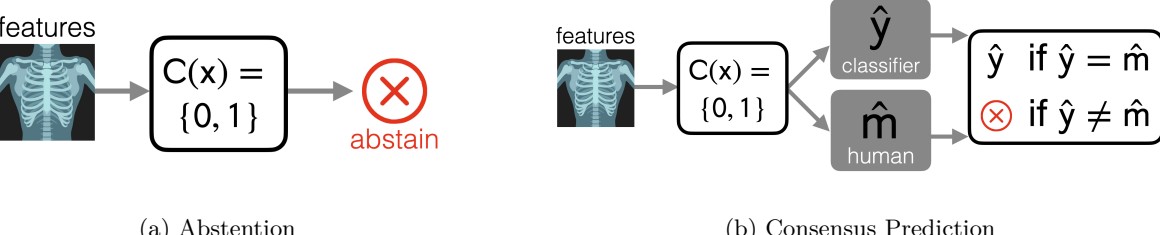

(a) Abstention

(b) Consensus Prediction

Figure 1: *L2D Decision Making Workflows.* Subfigure (a) shows that the L2D system will *abstain* if the rejector is uncertain. Subfigure (b) shows the alternative workflow in which, if the rejector is uncertain, the system will check for *consensus* between the expert and model predictions and abstain otherwise.

covariate shift caused by corruption noise (Ovadia et al., 2019). Even low-levels of noise can cause modern classifiers to start to fail, but a human can often be robust to similar noise. Another example of such human superiority is the case of adversarial examples, which by definition do not fool humans but fool a classifier. To cover such cases in which safety concerns dictate that the human should have priority, we call our third workflow 'human-preferred prediction. This means that if $C_r(\boldsymbol{x}_{N+1}; \hat{\tau}) = \{0, 1\}$, the L2D system will still query and return the human's prediction as the final output, despite the uncertainty. In this case, we are not using the deferral set in the same way as the previous two workflows since covariate shift is assumed to be happening. This shift violates the core assumptions of CP, invalidating the coverage guarantee.

### 3.3 Constructing and Using Deferral Intervals

Rather than producing a prediction set for the binary classifier (Section 2.2), we can instead consider an uncertainty interval for the rejector's confidence itself. This would result in the uncertainty interval of the form $[b_l(\mathbf{x}), b_r(\mathbf{x})]$, such that the conditional endpoints satisfy $b_l(\mathbf{x}) \in [0, 1)$, $b_r(\mathbf{x}) \in (0, 1]$, and $b_l(\mathbf{x}) \leq b_r(\mathbf{x})$. The coverage guarantee would then be $\mathbb{P}(\mathbb{P}(\mathrm{m} = \mathrm{y}|\mathbf{x}) \in [b_l(\mathbf{x}), b_r(\mathbf{x})]) \geq 1 - \alpha$, where $\alpha \in [0, 1]$ again controls the nominal coverage. Barber (2020) provides an algorithm to compute this interval in practice. Their algorithm can be applied straightforwardly to this case.

Moving on to decision-making, the deferral interval can be used the same way as the deferral set, as outlined in Section 3.2. The only change is that instead of, for example, abstaining when the set is of size two, here we need to specify a certain width that, when exceeded, triggers abstention. One possibility is to simply abstain if the interval is on both sides of 50% (i.e. $0.5 \in [b_l(\mathbf{x}), b_r(\mathbf{x})]$) and to not abstain when the interval is contained on one side (i.e. $0.5 < b_l(\mathbf{x})$ or $0.5 > b_r(\mathbf{x})$).

Yet unlike with deferral sets, this interval allows for a deferral decision that is close to traditional L2D but can be made more robust. Instead of just comparing the point estimates of the rejector and classifier confidences, we can use the interval and only defer when we are very sure that the human is more likely to be correct than the classifier: `defer` if $b_l(\mathbf{x}) > \max_{y \in \mathcal{Y}} p(\mathrm{y} = y|\mathbf{x})$. Doing so will ostensibly save in expert queries since the system will only call the expert when they clearly improves upon the classifier.

**Model-Preferred Prediction for Cost Saving**  While the abstention and consensus prediction workflows in Section 3.2 apply likewise to deferral intervals, we propose an additional workflow, model-preferred prediction. When the uncertainty interval is $[0, 1]$, we are unsure if the expert will be correct, and since experts usually require some expense to query, then we may want to query the expert only when we are sure they will be correct. Otherwise, the L2D system may have just as an acceptable an outcome querying the model, which often requires a negligible cost to query. We expect this workflow to increase the classifier's coverage while not substantially decreasing overall system accuracy.

In preliminary experiments, we found deferral intervals improved allocation calibration (ECE $5.86 \rightarrow 5.17$). However, the intervals were often too wide to support informative routing decisions at the desired coverage level. This reflects a known calibration–efficiency trade-off in conformal inference: marginal coverage is enforced by inflating interval size (Angelopoulos & Bates, 2023; Barber et al., 2023), especially with split

calibration and limited signal, which can yield intervals that are decision-agnostic (e.g., near $[0, 1]$) and thus uninformative for cost-aware thresholds. We therefore defer these results to Appendix C.

## 4 Related Work

The L2D framework (Madras et al., 2018) along with its precursors (Chow, 1957; Bartlett & Wegkamp, 2008; Yuan & Wegkamp, 2010; Cortes et al., 2016b) have received much attention of late due to their potential to improve safety via semi-automation (Raghu et al., 2019). The majority of such attention has focused on L2D's learning objective (Mozannar & Sontag, 2020; Verma & Nalisnick, 2022; Mao et al., 2024c;b; Cao et al., 2023; Charusaie et al., 2022) and its extension to multiple experts (Verma et al., 2023; Mao et al., 2024a; Keswani et al., 2021; Hemmer et al., 2022). Only two works have previously considered the uncertainty estimation abilities of the rejector sub-component, with Verma & Nalisnick (2022) first observing the aforementioned pathologies of the symmetric softmax parameterization and Cao et al. (2023) proposing the asymmetric softmax as a remedy. Liu et al. (2022) employed ensembling to estimate the classifier's uncertainty and used this to inform the deferral decision, but their approach does not model the expert's abilities nor represent the expert's uncertainty.

**Conformal Prediction for L2D** CP has previously been incorporated into L2D and related frameworks. Straitouri et al. (2023) and Babbar et al. (2022) both proposed performing CP for a classifier and then passing the set to a human to choose the label that will be the final prediction. Yet like the aforementioned approach by Liu et al. (2022), the classifier's uncertainty is being quantified, not the human's, which is the focus of our methodology. The work of Verma et al. (2023) is more related: they apply CP to multi-expert L2D to quantify the uncertainty in who is the *best* expert of the multiple available. Their coverage guarantee is formulated with the goal of including this best expert in the set. Our approach could be applied to multi-expert L2D, but it would construct a deferral set per expert, not across experts as they do.

**Large Language Models using Conformal Prediction for Deferral** With the rapid adoption of Large Language Models (LLMs), conformal prediction has become a vital tool for safety, alignment and model routing. Several concurrent approaches design conformal prediction with LLMs to mitigate hallucinations on various tasks (Quach et al., 2024; Cherian et al., 2024; Huang et al., 2025). Su et al. (2025) introduced an uncertainty-aware router that defers hard samples from a weak LLM to a strong reasoning model based on conformal set sizes. However, this work is not done for the L2D framework.

## 5 Experiments

We now experimentally demonstrate that incorporating uncertainty via CP into the deferral decision can have tangible benefits to the safety and robustness of L2D systems. Our experiments follow closely the setup in previous works on L2D (Mozannar et al., 2023; Verma et al., 2023; Cao et al., 2023) while introducing uncertainty quantification for the rejector. We trained L2D models using the OvA and A-SM surrogate losses. Taking this base L2D model, we then apply the CP procedure described in Section 3. Our implementations are publicly available at `https://github.com/yizirui/conformal_L2D`. See the supplementary materials for additional details, including training hyperparameters and backbone architectures.

**Datasets** We utilize three datasets tailored to different tasks: `CIFAR-10` (Krizhevsky, 2009) for image classification, `HAM10000` (Tschandl et al., 2018) for skin lesion diagnosis, and `Hate Speech` (Davidson et al., 2017) for hate speech detection. The `CIFAR-10` dataset comprises 60,000 instances, divided into training, calibration, and test sets at 70%, 10%, and 20%, respectively. Similarly, `Hate Speech` contains 25,000 instances, split into the same proportions. The `HAM10000` with 10,015 dermatoscopic images, is divided into 60% training, 20% calibration, and 20% test splits.

**Models and Experts** We follow previous work's L2D experimental settings (Mozannar et al., 2023; Verma & Nalisnick, 2022; Verma et al., 2023), including their choice of base model backbones and expert simulations. We apply a three-layer convolutional neural network (CNN) for `CIFAR-10`, a 34-layer residual

network (ResNet34) for `HAM10000`, and a linear network and SBERT embedding (Reimers & Gurevych, 2019) for `Hate Speech`. Expert simulations mirror Mozannar et al. (2023): on `CIFAR-10`, an oracle predicts perfectly on the first $k=6$ classes and uniformly at random on the remaining $(10-k)$; on `Hate Speech`, we use a stochastic "random-annotator" baseline; on `HAM10000`, an MLP-Mixer trained on meta-data emulates an expert with contextual information beyond pixels (Tolstikhin et al., 2021).

Table 1: *Coverage and Efficiency.* We report mean and standard deviation of the empirical coverage and the average size of the deferral set for a confidence level of $1 - \alpha = 90\%$.

| Dataset | Parameterization | Coverage (%) | Average Set Size |
|---|---|---|---|
| CIFAR-10 | OvA | 86.94 $\pm$ 0.86 | 1.07 $\pm$ 0.03 |
| | A-SM | 90.53 $\pm$ 0.56 | 1.37 $\pm$ 0.01 |
| HAM10000 | OvA | 90.65 $\pm$ 0.63 | 1.25 $\pm$ 0.01 |
| | A-SM | 91.13 $\pm$ 0.58 | 1.28 $\pm$ 0.03 |
| HateSpeech | OvA | 90.35 $\pm$ 0.53 | 1.03 $\pm$ 0.03 |
| | A-SM | 90.67 $\pm$ 0.52 | 1.01 $\pm$ 0.01 |

## 5.1 Coverage and Efficiency

We first experimentally verify that the target coverage is met, validating CP's guarantee (Equation 7). In Table 1, we report the empirical coverage and average set size for the three aforementioned datasets. Both parameterizations meet the target coverage level (90%) for all datasets except for OvA on `CIFAR-10` ($\sim 87\%$). In all cases, the sets are quite efficient, with the average set size always being less than 1.3. The exceptionally small set size of 1.07 for OvA on `CIFAR-10` leads to its mis-coverage. We suspect the mis-coverage is due to (natural) train-test distribution shift.

## 5.2 Learning to Defer with Abstention and Consensus Prediction

We next investigate the efficacy of the abstention and consensus decision-making workflows presented in Section 3.2. Table 2 reports the system accuracy, ratio of test points deferred, and the coverage of the system (i.e. the fraction of points for which the system does not abstain) again for `CIFAR-10`, `Hate Speech`, and `HAM10000`. We see that both OvA and A-SM improve upon the system accuracy of the base L2D model for `CIFAR-10` and `HAM10000`, with improvements ranging from 2 to 5 percentage points (in absolute terms). However, the coverage reduction is variable, ranging from modest ($-8$ percentage points) to substantial ($-38$ percentage points), meaning that the system accuracy improvement would be practical in some cases (e.g. OvA for `CIFAR-10`) and not in others (e.g. A-SM for `CIFAR-10`). On `Hate Speech`, abstention occurred for very few points, leading to uninteresting system accuracy results. We do not see a clear superiority between the parameterizations.

Table 2 highlights the 'safety-availability' trade-off inherent to conformal L2D. This drop implies a 'routing ambiguity' where the rejector cannot confidently distinguish between expert and model capability. The consensus prediction workflow demonstrates how we can recover some of this lost coverage. By querying both model and human when the rejector is uncertain, we increase coverage significantly (e.g., 67.57% for `CIFAR-10` A-SM) while maintaining comparable system accuracy.

## 5.3 Learning to Defer under Covariate Shift

To evaluate three workflows under covariate shift, we induce out-of-distribution (OOD) shift with a six-level severity index (1–6), where levels 1–5 increase smoothly. We then define level 6 as the *extreme-shift* condition across all datasets. The transition from level 5 to 6 represents a markedly larger distributional change than the preceding increments. On `CIFAR-10`, we utilized the brightness corruption subset of `CIFAR-10-C` for severity levels 1 to 5 and use `SVHN` at level 6 to induce a semantic shift. For `HAM10000`, we compose two image corruptions with severity-controlled parameters: (i) masking with fraction $p$ growing from 10% to

Table 2: *Abstention and Consensus Prediction.* We report mean and standard deviation of system accuracy, fraction of points deferred, and test-set coverage.

| | Parameterization | Method | System Accuracy | Fraction Deferred | System Coverage |
|---|---|---|---|---|---|
| **CIFAR-10** | OvA | Base Model | 84.71 ± 0.46 | 55.26 ± 1.76 | 100 |
| | | Abstention | 86.72 ± 1.02 | 56.41 ± 2.30 | 92.14 ± 0.48 |
| | | Consensus | **86.79** ± 1.07 | 56.38 ± 2.31 | 93.32 ± 0.52 |
| | A-SM | Base Model | 84.01 ± 0.45 | 56.63 ± 3.73 | 100 |
| | | Abstention | 87.05 ± 0.76 | 84.13 ± 4.56 | 62.53 ± 0.75 |
| | | Consensus | **87.58** ± 0.61 | 79.62 ± 4.31 | 67.57 ± 0.75 |
| **HAM10000** | OvA | Base Model | 82.1 ± 0.49 | 33.71 ± 2.39 | 100 |
| | | Abstention | **87.48** ± 0.51 | 35.91 ± 2.84 | 75.23 ± 1.40 |
| | | Consensus | 85.72 ± 0.63 | 34.27 ± 2.52 | 88.39 ± 1.85 |
| | A-SM | Base Model | 78.92 ± 0.29 | 26.68 ± 3.07 | 100 |
| | | Abstention | **87.05** ± 0.87 | 28.11 ± 3.45 | 72.82 ± 1.19 |
| | | Consensus | 84.76 ± 0.44 | 27.49 ± 3.16 | 84.48 ± 0.95 |
| **Hate Speech** | OvA | Base Model | 92.09 ± 0.07 | 42.41 ± 0.99 | 100 |
| | | Abstention | **92.28** ± 0.14 | 42.48 ± 0.96 | 99.38 ± 0.43 |
| | | Consensus | 92.25 ± 0.13 | 42.42 ± 0.96 | 99.78 ± 0.22 |
| | A-SM | Base Model | 91.82 ± 0.32 | 67.91 ± 1.76 | 100 |
| | | Abstention | **91.88** ± 0.15 | 67.79 ± 1.74 | 99.16 ± 0.75 |
| | | Consensus | **91.88** ± 0.12 | 67.81 ± 1.73 | 99.65 ± 0.28 |

50% at level 5 and 80% at level 6, and (ii) Gaussian blur with kernel size increasing from 3×3 to 13×13 and their standard deviation $\sigma$ from 0 to 4 at level 5 then 6 at level 6. For `HateSpeech`, we perturb embeddings with adversarial noise injection (Wei & Zou, 2019; Donahue et al., 2017): progressively from level 1 to 5 (i) dimension masking with ratio $r \in [5\%, 25\%]$, (ii) additive jitter with standard deviation increasing from $\approx 2\%$ to $\approx 10\%$, and (iii) simple token-inspired edits with rates up to 25%. Level 6 constitutes an extreme step: $r = 40\%$, jitter $\approx 15\%$, and edit rates $\approx 40\%$. Here, we include additional baselines that test other uncertainty methods (deep ensembling) and if the held-out data we use for split-CP could be better used for finetuning. We detail these additional method variants below.

**Original** & **Original A-SM** We built our original models with OvA surrogate (Verma et al., 2023) and with A-SM surrogate (Cao et al., 2023) on dataset without shift.

**Conformal** Following Section 3, we calibrate the rejector with split-CP on a calibration set from the non-shifted dataset to compute the threshold $\hat{\tau}$.

**Conformal + Shifted $D_{\mathbf{cal}}$** While the above conformal method does not fit to the target distribution, a subset from the target distribution is expected to calibrate the system to target distribution in computing $\tau$ in Section 3.

**Finetuned on Shift (Baseline)** We perform a finetuning to see how L2D framework could take the most advantage of the limited and invaluable data from target distribution. Rejectors were trained on the source distribution will apply on the same subset in the Conformal + Shifted $D_{\mathrm{cal}}$ method to finetune.

**Selective Prediction (Baseline)** To disentangle the benefits of the L2D rejector from simple uncertainty thresholding, we compare against a Selective Prediction baseline (Mozannar et al., 2023; Geifman & El-Yaniv, 2017). This method relies solely on the base classifier's vanilla confidence, defined as the maximum softmax probability. A threshold $\tau$ is tuned on the calibration set to maximize system accuracy: prediction is deferred if the classifier's confidence is below $\tau$, and retained otherwise.

**Ensemble (Baseline)** We approached the uncertainty ensemble by Liu et al. (2022) in a computationally efficient way by taking advantage of the already trained L2D functions. An uncertainty ensemble can be constructed by reinitializing and trivially retraining $m$ rejector layer functions based on above trained L2D functions. To align with the size of the conformal set $C_r(\mathbf{x}; \hat{\tau})$ in this study, we set $m = 2$.

**Prediction** Figure 2 reports the system accuracy and ratios of deferred instances in the population of the abstention, consensus, human-preferred methods shown in Figs. 2d, 2h and 2l respectively for OOD data. In this scenario, it is notable that under a non-extreme distribution shift, both proposed conformal methods

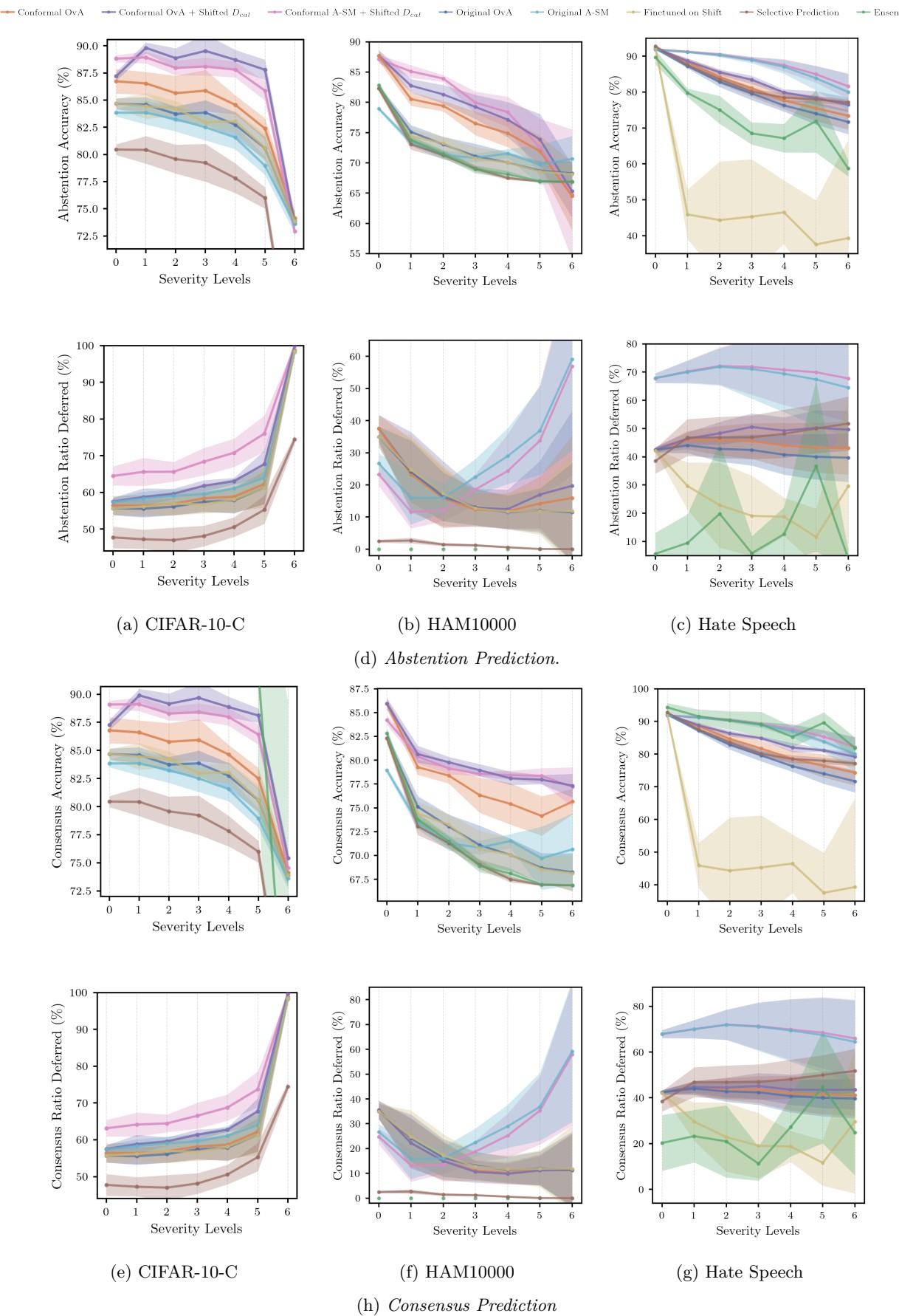

(a) CIFAR-10-C     (b) HAM10000     (c) Hate Speech

(d) *Abstention Prediction.*

(e) CIFAR-10-C     (f) HAM10000     (g) Hate Speech

(h) *Consensus Prediction*

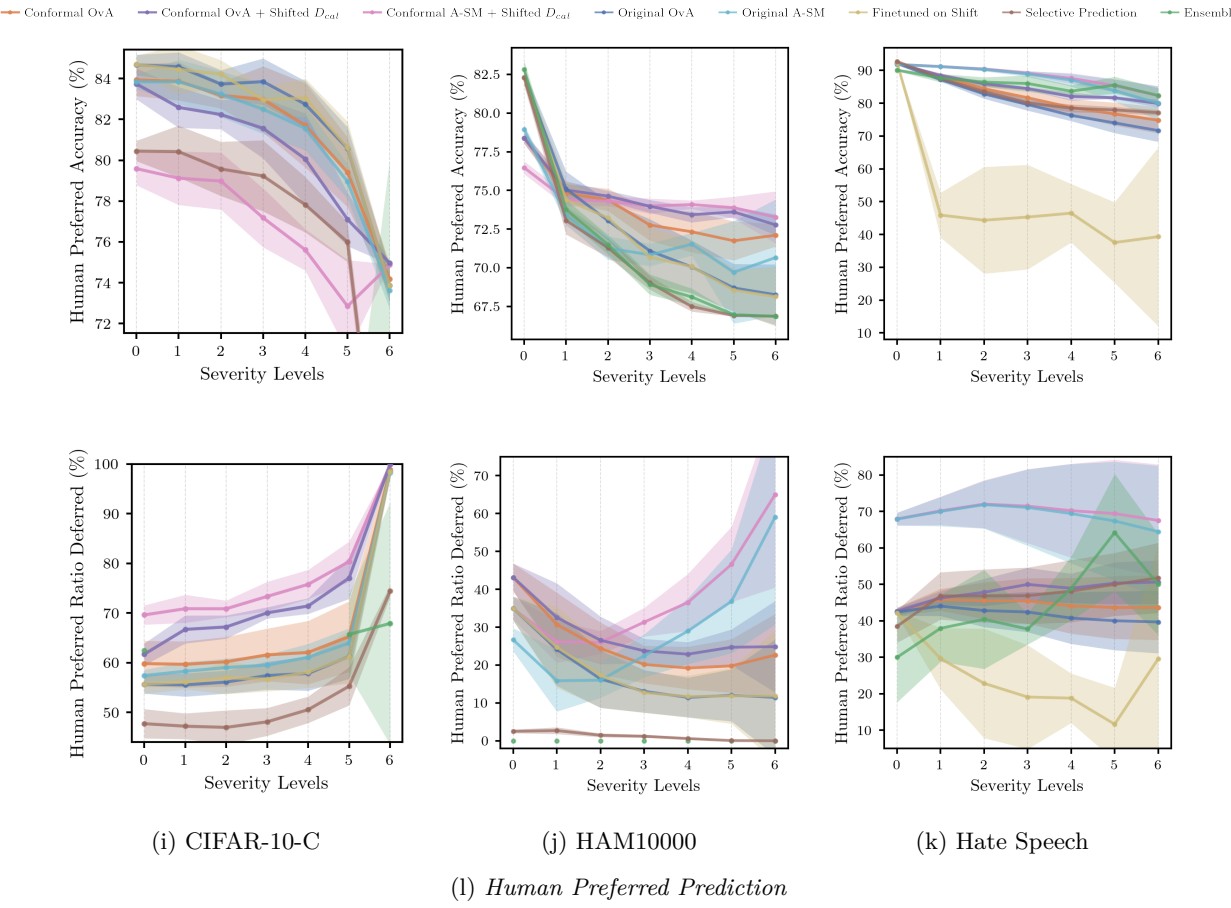

(i) CIFAR-10-C          (j) HAM10000          (k) Hate Speech

(l) *Human Preferred Prediction*

Figure 2: *L2D via Conformal Prediction on OOD.* Figures above report the mean and standard deviation of system accuracy and ratio of deferred instances for methods stated in this section via abstention, consensus, and human-preferred prediction under different levels of distribution shift for `CIFAR-10C`, `HAM10000`, and `Hate Speech`. The proposed L2D framework, employing both OvA and A-SM surrogate losses, exhibits less fluctuation and greater robustness against covariate shift compared to most listed methods, although model deterioration remains evident.

outperform the original L2D framework and selective prediction. In particular, *Conformal OvA + Shifted $D_{cal}$* and *Conformal A-SM + Shifted $D_{cal}$* methods demonstrate effective deferral behavior by recognizing uncertainty in the test data, thereby maintaining the overall robustness and accuracy of the system. This approach limits performance degradation across shifts from levels 1 to 5 in `CIFAR-10` and `HAM10000` from level 1 to 5. On the `Hate Speech`, the performance decline was mitigated, showing a slower rate of deterioration.

**Discussion** The divergent behaviors observed in Figure 2 underscore the necessity of uncertainty quantification in the deferral decision under shift. As evidenced by the rising deferral ratios in Figure 2, the *Conformal + Shifted $D_{cal}$* methods dynamically adapt to the shift intensity. For instance, in HAM10000 levels 4-6, *Conformal OvA + Shifted $D_{cal}$* move from 75.91% to 74.97% while *Original OvA* move from 69.01% to 67.00%. This suggests that recalibrating the rejector on even a small sample, for example 10% calibration compared with 70% of training or 20% of testing, from the target distribution allows the system to effectively detect when the model's competence region has been exited. Consequently, the system preserves overall reliability by routing ambiguous instances to the consensus or the human expert or abstaining entirely.

## 5.4 Evaluation of Rejector Uncertainty

In Section 3.3, we incorporate uncertainty interval to defer and abstain conservatively. Specifically, a L2D system should not defer when a human expert is not believed to predict correctly under confidence level $1 - \alpha$. We introduce two metrics that jointly characterize the performance of the rejector when deferring: uncertainty rejector accuracy $a\tilde{c}c(m = y)$ and uncertainty classifier accuracy $a\tilde{c}c(y = y)$ as follows:

$$\widetilde{\mathrm{acc}}_{m=y} = \frac{\sum \mathbb{I}[m = \mathrm{y}, r(\boldsymbol{x}) = 1]}{\sum \mathbb{I}[r(\boldsymbol{x}) = 1]}, \quad \widetilde{\mathrm{acc}}_{\mathrm{y}=y} = \frac{\sum \mathbb{I}[\hat{y} = \mathrm{y}, r(\boldsymbol{x}) = 0]}{\sum \mathbb{I}[r(\boldsymbol{x}) = 0]},$$

Here, $r(\mathbf{x}) = 1$ indicates deferral, and $r(\mathbf{x}) = 0$ indicates acceptance of the classifier's prediction. The metric $\widetilde{\mathrm{acc}}_{m=y}$ measures the correctness of deferred predictions—assuming the original model $m$ approximates the human expert, and $\widetilde{\mathrm{acc}}_{\mathrm{y}=y}$ quantifies the accuracy of predictions made autonomously. Together, these metrics summarize a L2D system's decision-making performance under uncertainty. We observed improvements in both metrics under L2D with deferral interval, particularly large in gains $\widetilde{\mathrm{acc}}_{\mathrm{y}=y}$; detailed results appear in Appendix C. These gains indicate that uncertain instances are predominantly deferred.

## 6 Conclusions, Limitations, and Future Work

We applied conformal prediction to the rejector component of the learning-to-defer framework with both one-vs-all and asymmetric softmax parameterizations. This approach offers finite-sample, distribution-free guarantees for quantifying uncertainty in the expert's predictions. Our experiments demonstrate that not only does our method achieve the targeted coverage guarantees with compact prediction sets, but the resulting deferral sets or intervals also enable alternative decision-making workflows, such as abstention or expert–model consensus. In particular, we advocate an abstention workflow that empowers the system to respond "I don't know who knows" when its confidence is insufficient, thereby enhancing the safety and robustness of human–AI collaboration. Such a system alerts users to gather additional information before making a confident decision.

The primary limitation of our work is that our deferral sets are constructed based on expert correctness, not on the comparison that truly dictates the deferral decision. They reflect but do not perfectly align with the Bayes optimal rejector that compares the expert's correctness probability with the classifier's confidence. Thus extending our CP procedure to somehow fuse and compare the uncertainty in the classifier and rejector is an exciting and impactful direction for future work that would address this limitation. Another direction is to apply distribution-free risk control (Angelopoulos et al., 2024) to ensure the deferral decision respects certain constraints, such as bounding the deferral rate.

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

# A  Theoretical Supplement

## A.1  Expected size of the deferral set under covariate shift

The widening of deferral sets under covariate shift acts as a functional safety mechanism. This behavior arises from the specific construction of our deferral set $C_r(x; \hat{\tau})$ in Eq. 8. The rejector outputs the uncertain set $\{0, 1\}$ if and only if the expert correctness probability satisfies: $\hat{p}(\mathrm{m} = \mathbf{y}|\mathbf{x}) \in \mathcal{U}$. On in-distribution data, the rejector is confident, and the density of $\hat{p}(\mathrm{m} = \mathbf{y}|\mathbf{x})$ concentrates at the boundaries $\{0, 1\}$, minimizing the frequency of uncertain sets. However, as the rejector faces out-of-distribution inputs, its predictive performance degrades thus tends to produce less distinguishable probabilities on 0 and 1, i.e. moving $\hat{p}(\mathrm{m} = \mathrm{y} \mid \mathbf{x})$ closer to 0.5, where the system is roughly 50–50 about whether the expert will be correct. As the score distribution concentrates within $\mathcal{U}$, the threshold $1 - \hat{\tau}$ is more often not satisfied to produce a singleton deferral set, so deferral sets of size two become more frequent precisely in those regions.

This phenomenon is formalized in the below section. To simplify, let $\hat{p}(\mathbf{x}) = \hat{p}(\mathrm{m} = \mathbf{y} \mid \mathbf{x})$ be the rejector's estimate of the expert-correctness probability, and let $C_r(\mathbf{x}; \hat{\tau})$ be the conformal deferral set defined in Eq. 8. The conformal deferral set satisfies $|C_r(\mathbf{x}; \hat{\tau})| \in \{1, 2\}$. We can have

$$|C_r(\mathbf{x}; \hat{\tau})| = \begin{cases} 1, & \hat{p}(\mathbf{x}) \leq \hat{\tau} \text{ or } \hat{p}(\mathbf{x}) \geq 1 - \hat{\tau}, \\ 2, & \hat{\tau} < \hat{p}(\mathbf{x}) < 1 - \hat{\tau}. \end{cases}$$

and hence

$$|C_r(\mathbf{x}; \hat{\tau})| = 1 + \mathbb{I}\{\hat{\tau} < \hat{p}(\mathbf{x}) < 1 - \hat{\tau}\}.$$

Therefore, for any covariate distribution $Q_{\mathbf{x}}$,

$$\mathbb{E}_{\mathbf{x} \sim Q_{\mathbf{x}}}\big[|C_r(\mathbf{x}; \hat{\tau})|\big] = 1 + Q_{\mathbf{x}}(\hat{p}(\mathbf{x}) \in \mathcal{U}).$$

This identity shows that deferral sets become wider precisely when a larger fraction of test points fall into the middle-confidence region $\mathcal{U}$. To express this more symmetrically around the decision boundary 0.5, define a probability margin

$$\gamma_p(x) := |2\hat{p}(\mathbf{x}) - 1| \in [0, 1].$$

Assuming $\hat{\tau} \leq 0.5$, the event $\hat{p}(\mathbf{x}) \in (\hat{\tau}, 1 - \hat{\tau})$ is equivalent to $\gamma_p(\mathbf{x}) < 1 - 2\hat{\tau}$. Hence,

$$\mathbb{E}_{\mathbf{x} \sim Q_{\mathbf{x}}}\big[|C_r(\mathbf{x}; \hat{\tau})|\big] = 1 + Q_{\mathbf{x}}(\gamma_p(\mathbf{x}) < 1 - 2\hat{\tau}).$$

A sufficient condition for wider deferral sets under a shifted covariate distribution $Q_{\mathbf{x}}$ with relative to the source distribution $P_{\mathbf{x}}$ is that more shifted points have small margin, i.e.,

$$Q_{\mathbf{x}}(\gamma_p(\mathbf{x}) < 1 - 2\hat{\tau}) \ \geq \ P_{\mathbf{x}}(\gamma_p(\mathbf{x}) < 1 - 2\hat{\tau}).$$

Under this condition, the expectation follows that

$$\mathbb{E}_{\mathbf{x} \sim Q_{\mathbf{x}}}\big[|C_r(\mathbf{x}; \hat{\tau})|\big] \ \geq \ \mathbb{E}_{\mathbf{x} \sim P_{\mathbf{x}}}\big[|C_r(\mathbf{x}; \hat{\tau})|\big].$$

This presents, when distribution shift makes the rejector less separable between $\mathrm{m} = \mathbf{y}$ and $\mathrm{m} \neq \mathbf{y}$ (equivalently, $\hat{p}(\mathbf{x})$ is closer to 0.5, so $\gamma_p(x)$ shrinks), more points fall into $\mathcal{U}$, increasing the expected deferral set size. We emphasize that this is a conditional explanation of the empirical widening behavior, and does not by itself provide a coverage guarantee under arbitrary distribution shift.

## A.2  Theoretical Formalization of Human-Preferred and Model-Preferred Prediction

**Human-Preferred Workflow**   The Human-Preferred Workflow is designed for scenarios where the test distribution $P_{test}(\mathbf{x}, y)$ differs from training $P_{train}(\mathbf{x}, y)$.

**Assumption 1 (Human Robustness):** While machine classifiers suffer significant performance degradation under distribution shifts (e.g., image corruptions, adversarial attacks), it is well-established that human

perception remains comparatively invariant. Geirhos et al. (2019) demonstrate that CNNs rely on texture bias which is fragile to shift, while humans rely on shape bias which is robust. Dodge & Karam (2017) show human vision significantly outperforms DNNs on distorted images. Humans are also robust to adversarial perturbations that fool models (Zhou & Firestone, 2019).

In addition, the human expert can access privileged metadata that is unavailable to the model. For example, in our HAM10000 experiments, the expert utilizes metadata beyond pixel information. When the conformal set is $C_r(\mathbf{x}) = \{0, 1\}$, the rejector indicates high uncertainty about the standard inductive relationship between input and correctness. Under distribution shift, this uncertainty signal is correlated with the model entering a failure mode. Let $R(h)$ and $R(\mathrm{m})$ be the risk of the classifier and human, respectively. Under shift, we assume the inequality $R_{shift}(h) \gg R_{shift}(\mathrm{m})$. Therefore, the Human-Preferred workflow is formalized as a robust fallback policy.

$$
r^*(\mathbf{x}) = \begin{cases} 0, & \text{if } C_r(\mathbf{x}; \hat{\tau}) = \{0, 1\} \ \wedge \ \hat{\mathbf{y}} = \mathrm{m}, \ (\text{Consensus while Uncertainty detected}) \\ 1, & \text{if } C_r(\mathbf{x}; \hat{\tau}) = \{0, 1\} \ \wedge \ \hat{\mathbf{y}} \neq \mathrm{m}, \ (\text{Uncertainty detected}) \\ r(\mathbf{x}), & \text{otherwise.} \end{cases}
$$

This minimizes the expected risk upper bound by deferring to the agent with the lower variance in performance across distributions (the human), utilizing the size of the conformal set as a proxy for detecting out-of-distribution (OOD) samples.

**Model-Preferred Workflow** This workflow can be motivated through a cost-sensitive objective, where querying the human expert incurs a non-negligible cost while using the model is essentially free. Let $c_h > 0$ be the cost of consulting the human expert per query. A standard cost-sensitive system loss to minimize is:

$$
L_{0-1,\text{model-preferred}}(h, r) = \mathbb{E}\Big[(1 - r(\mathbf{x})) \cdot \mathbb{I}[h(\mathbf{x}) \neq \mathrm{y}] + r(\mathbf{x}) \cdot \mathbb{I}[\mathrm{m} \neq \mathrm{y}] + r(\mathbf{x}) \cdot c_h\Big].
$$

When the conformal deferral set is $C_r(\mathbf{x}; \hat{\tau}) = \{0, 1\}$, the rejector does not provide directional evidence at confidence level $1 - \alpha$ that deferring is beneficial to justify paying the query cost. Since the cost $c_h$ is incurred whenever $r(\mathbf{x}) = 1$, a budget-aware policy is to defer only when there is a confidently certified benefit, and otherwise default to the model. In particular, under the model-preferred workflow we set $r(\mathbf{x}) = 0$ whenever $C_r(\mathbf{x}; \hat{\tau}) = \{0, 1\}$ to avoid paying $c_h$ when uncertain.

## B Experimental Details

### B.1 Accuracy vs Deferred Ratio

Figure 3 summarizes the accuracy-coverage comparison across all datasets (CIFAR-10, HAM10000, and HateSpeech) and deferral workflow (abstention, consensus, and human-preferred).

### B.2 Datasets in Experiments

Table 3 summarizes the datasets used in this work. The `CIFAR-10` dataset consists of 60,000 32x32 color images, evenly distributed across 10 distinct classes (Krizhevsky, 2009). It is divided into 50,000 training images and 10,000 test images. The `CIFAR10-C` dataset introduces 15 common real-world corruptions, along with 4 additional types of corruption, applied to the test images from CIFAR-10. The `HAM10000` dataset (Tschandl et al., 2018) contains 10,015 dermatoscopic images, categorized into seven types of skin lesions: melanocytic nevi, melanoma, benign keratosis-like lesions, basal cell carcinoma, actinic keratoses, vascular lesions, and dermatofibroma. The `HateSpeech` dataset (Davidson et al., 2017) includes 24,802 labeled tweets, sampled from a total of 85.4 million tweets across 33,458 users, with labels divided into 3 categories.

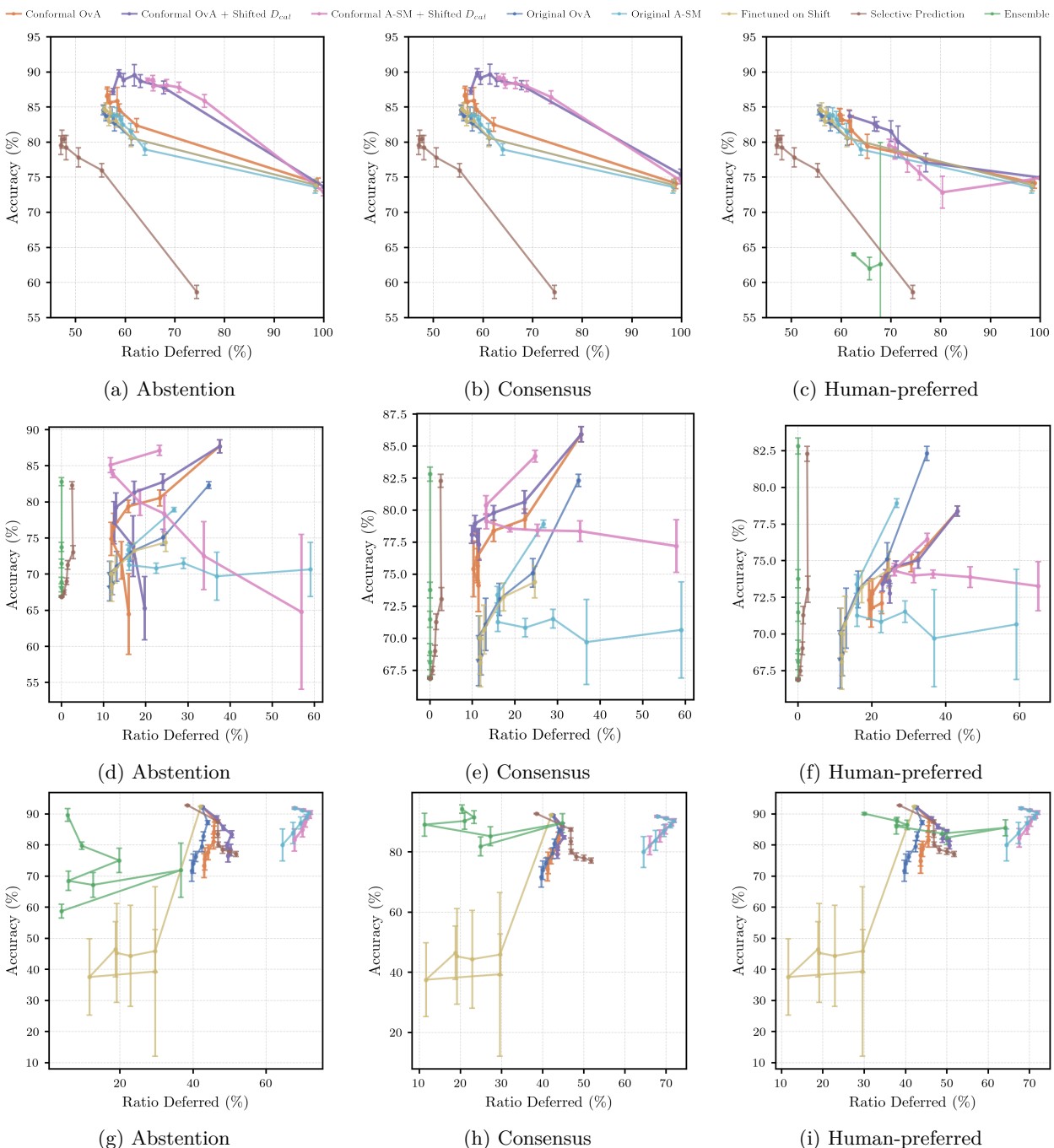

Figure 3: Accuracy–coverage for all datasets and deferral workflows. Each panel plots non-abstention accuracy versus the fraction of examples deferred to the human (ratio deferred). The top row corresponds to CIFAR-10, the middle row to HAM10000, and the bottom row to HateSpeech; within each row, the three columns show the abstention, consensus, and human-preferred deferral workflows, respectively.

### B.3   Distribution Shift Simulation

To simulate covariate shift in the HAM10000 dataset, we applied Gaussian blur and noise corruption to the images. Table 4 outlines the Gaussian blur parameters for varying levels of severity, with increasing kernel sizes and $\sigma$ values to progressively enhance the blurriness as the shift intensifies. To introduce covariate

| Dataset | $n$ | $K$ | Human Expert | Base Model | Optimizer | Epochs | Learning Rate |
|---|---|---|---|---|---|---|---|
| CIFAR-10 | 60k | 10 | synthetic $k$ classes | CNN with 2 Conv layers (width = 50) | Adam | 100 | $1 \times 10^{-3}$ |
| HAM10000 | 10015 | 7 | MLP Mixer | ResNet34 | SGD | 150 | $1 \times 10^{-3}$ |
| HateSpeech | 25k | 3 | random annotator | Linear Network with SBERT | Adam | 50 | $1 \times 10^{-2}$ |

Table 3: *Summary of datasets and model configurations used in this work.* Here, $n$ refers to the total number of samples in each dataset, $K$ represents the number of classes, epochs refers to the total training epochs for the L2D framework with a rejector, and base model indicates the foundational model used within the L2D framework.

| Severity Level | Kernel Size | $\sigma$ |
|---|---|---|
| 1 | (3, 3) | 0.5 |
| 2 | (5, 5) | 1.0 |
| 3 | (7, 7) | 2.0 |
| 4 | (9, 9) | 3.0 |
| 5 | (11, 11) | 4.0 |
| 6 | (13, 13) | 5.0 |

Table 4: Gaussian Blur Kernel Sizes and $\sigma$ Values for Different Severity Levels

shift in the Hate Speech dataset, adversarial noise was applied. This transformation was constructed using techniques such as synonym replacement, random insertion, random deletion, and character swapping on the text embeddings.

### B.4  Additional Detail of Conformal Prediction

For the split-conformal prediction procedure described in Section 3.

**Score Function** : We utilize the probability of the expert correctness $\hat{p}(m = y|\mathbf{x})$ as defined in Eq. 5 (OvA) and Eq. 6 (A-SM).

**Quantile Calculation** : The threshold $\hat{\tau}$ is computed as the $\lceil (N_{cal} + 1)(1 - \alpha) \rceil / N_{cal}$ empirical quantile of the calibration scores, with a finite-sample correction.

### B.5  Rejector Entropy

### B.6  Coverage and Efficiency at Higher Target Coverage Levels

In the main paper Table 1, we reported coverage and efficiency results for the OvA rejector at a target coverage level of $1 - \alpha = 90\%$. This level is standard in conformal prediction and uncertainty quantification, where prediction sets are often calibrated to contain the true label with probability around 90% in order to balance statistical validity with the size of the output sets (Angelopoulos & Bates, 2023; Angelopoulos et al., 2021).

To demonstrate that our method is not tied to a single confidence level, we additionally evaluate the OvA rejector at higher target coverage levels $1 - \alpha \in \{95\%, 99\%\}$ on CIFAR-10, HAM10000, and HateSpeech. The empirical coverage and average deferral-set size for these settings are summarized in Table 6. As discussed in the main paper, all empirical coverage are slightly above the corresponding target levels, except a few cases falling slightly below, which is consistent with the coverage guarantees. The results exhibit the expected coverage-efficiency trade-off and confirm that our framework can be tuned to stricter coverage requirements without changing the conclusions.

Table 5: *Mean entropy of the rejector's estimated expert correctness $\hat{P}(\mathrm{m} = \mathbf{y}|\mathbf{x})$ under increasing distribution-shift severity.*

|  | Shift severity | | | | | | |
|---|---|---|---|---|---|---|---|
| **Dataset** | 0 | 1 | 2 | 3 | 4 | 5 | 6 |
| CIFAR-10 | 0.3122 | 0.3394 | 0.3691 | 0.3988 | 0.4185 | 0.4227 | 0.3204 |
| HAM10000 | 0.1280 | 0.1063 | 0.1374 | 0.1457 | 0.1944 | 0.2645 | 0.2736 |
| Hate Speech | 0.3078 | 0.3487 | 0.3650 | 0.3714 | 0.3776 | 0.3834 | 0.3896 |

Table 6: *Coverage and Efficiency of OvA Rejector at Higher Confidence Levels.* We report the empirical coverage and the average size of the deferral set for confidence levels $1 - \alpha = 95\%$ and $1 - \alpha = 99\%$.

|  | $1 - \alpha = 95\%$ | | $1 - \alpha = 99\%$ | |
|---|---|---|---|---|
| Dataset | Coverage (%) | Average Set Size | Coverage (%) | Average Set Size |
| CIFAR-10 | 93.08 | 1.46 | 98.41 | 1.76 |
| HAM10000 | 95.72 | 1.86 | 98.59 | 1.77 |
| HateSpeech | 95.14 | 1.32 | 99.09 | 1.80 |

### B.7 Evaluation on Uncertainty of Rejector $r(\mathbf{x})$ with Deferral Set

Section 5.4 introduced evaluation metrics for the rejector under uncertainty in addition to *system accuracy*: *uncertainty rejector accuracy* $\widetilde{acc}_{\mathrm{m}=y}$, and *uncertainty classifier accuracy* $\widetilde{acc}_{\mathrm{y}=y}$. While the main paper focuses on overall system accuracy and ratio deferred Figure 2, here we report these metrics across severity levels for the same methods considered in Section 5.3.

In particular, *Original* denotes the original OvA L2D model without conformal calibration, rows without the "+ Shifted $D_{cal}$" suffix (Abstention, Consensus, Human Preferred) correspond to the Conformal OvA rejector calibrated on the source distribution, and rows with "+ Shifted $D_{cal}$" correspond to Conformal OvA recalibrated on a small subset of the shifted target distribution.

## C Experiment: L2D with Deferral Intervals

This section documents the experiment evaluating L2D workflow with conformal deferral intervals, both without and with **Conformal + Shifted $D_{\mathbf{cal}}$** (i.e., recalibrating on a small target-domain subset under shift).

**Reliability Diagram** We perform evaluation of calibration and examine the validity of conformal prediction by plotting reliability diagram and computing expected calibration error (ECE). We define the expected accuracy and ECE as

$$\widehat{\mathrm{acc}}(c) = \mathbb{P}(\mathrm{m} = \mathbf{y}|p(\mathrm{m} = \mathrm{y}|\mathbf{x}) = c), \quad \mathrm{ECE}_{p(\mathrm{m}=\mathrm{y})} = \mathbb{E}_{\mathbf{x}} \left| \mathbb{P}(\mathrm{m} = \mathbf{y} \mid p(\mathrm{m} = \mathrm{y}|\mathbf{x}) = c) - c \right|,$$

where $c$ is the confidence level. From the prediction interval $[b_l(\mathbf{x}), b_r(\mathbf{x})]$, the uncertainty bar of the expected accuracy could be calculated as $\Delta_- = b_l(\mathbf{x}) \, \mathbb{I}[\mathrm{m} = \mathrm{y}], \Delta^+ = b_r(\mathbf{x}) \, \mathbb{I}[\mathrm{m} = \mathrm{y}]$. Figure 4 presents the reliability diagram and constructs the error bar by distributing the prediction interval across the accuracy.

**Uncertainty Rejector Evaluation** we evaluate the performance of the uncertainty rejector $r(\mathbf{x})$ by accuracy metrics $\widetilde{acc}_{m=y}$ and $\widetilde{acc}_{y=y}$ on `CIFAR-10` datasets. Table 10 shows a significant increase in L2D with the prediction interval in $\widetilde{acc}_{m=y}$ and $\widetilde{acc}_{y=y}$, while the deferral interval is too large. This confirms that the binary regression would adjust the rejector to make deferral decision and abstention with care of uncertainty. It also reports that with abstention, the system accuracy increases from 83.87% to 93.75%, Table 10 evaluated the $\widetilde{acc}_{m=y}$ and $\widetilde{acc}_{y=y}$ in L2D with deferral interval.

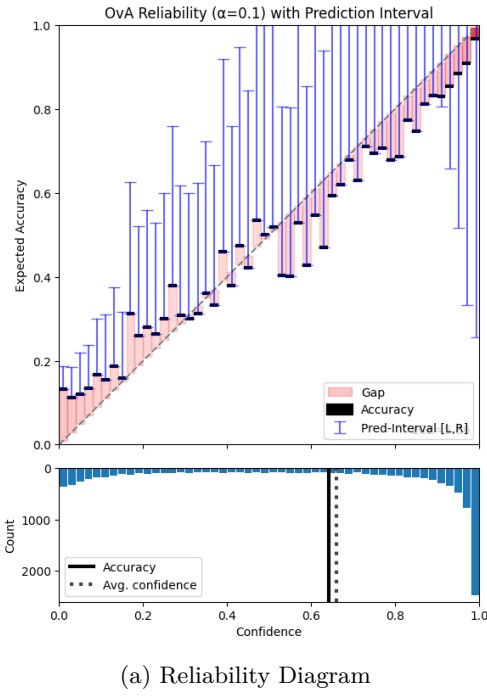
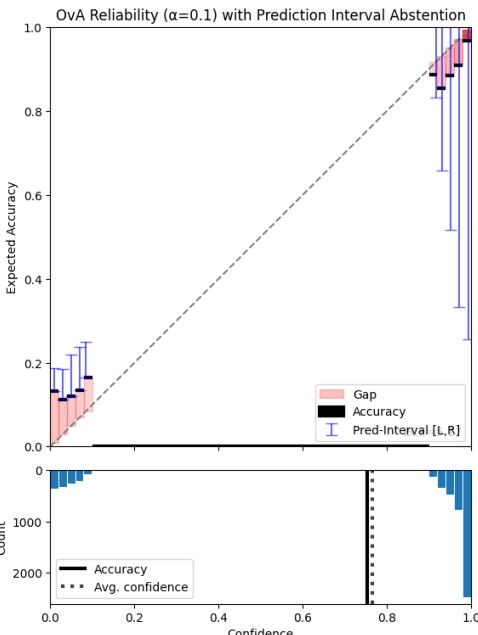

(a) Reliability Diagram

(b) Reliability Diagram with Abstention

Figure 4: Evaluation of Calibration on CIFAR-10: Subfigure (a) reports a reliability diagram w/o abstention and the expected calibration error (ECE) is 5.86. Subfigure (b) reports a reliability diagram w/ abstention and the ECE is 5.17

# D   Additional Experiment

## D.1   Dual CP: Conformal Prediction on Both Classifier and Expert Model

While Section 3 approaches optimal rejector with human expert correctness $\mathbb{I}\left[m_{N+1} = y_{N+1}\right]$. In this section, we present our exploration to compare the calibrated uncertainty of both the classifier and the expert model to make the deferral decision explicitly. To this end, we fit a weak classifier $human(\mathbf{x})$ to simulate the human expert predictions m from features $\mathbf{x}$, yielding a probabilistic predictor $\hat{p}_{human}(y \mid \mathbf{x})$. One may explicitly construct classifier prediction sets $C_h(\mathbf{x}; \hat{\tau}_h)$ for classifier $h^*(\mathbf{x})$ and annotator prediction sets $C_{human}(\mathbf{x}; \hat{\tau}_{human})$ for human expert $human^*(\mathbf{x})$ following the split conformal prediction construction in Section 2.2.

We then defer when the classifier is more uncertain by comparing the sizes of the two prediction sets under the same confidence level $1 - \alpha$:

$$r^*(\mathbf{x}) = \begin{cases} 1 \text{ if } |C_h(\mathbf{x}; \hat{\tau}_h)| > |C_{human}(\mathbf{x}; \hat{\tau}_{human})|, \\ 0 \text{ otherwise.} \end{cases}$$

We reuse the experimental setup and one-vs-all (OvA) parameterization from Section 5. We then apply the dual conformal-prediction (CP) procedure described above to the CIFAR-10, HAM10000, and Hate Speech datasets without shift. The resulting coverage and deferral-rate metrics are reported in Table 11.

## D.2   Finetuning on Uncertain Instances

To broaden coverage, one may finetune the L2D system on instances that CP flags as uncertain. The initial threshold $\hat{\tau}$ could be again computed over initial calibration set $\mathcal{D}_2$ as Section 3.1. Building on the dual-CP deferral setup in Appendix D.1, we now run CP on the training set $\mathcal{D}_1$ and form prediction set $C_r(\mathbf{x}; \hat{\tau})$.

We could then define the uncertain subset by a cardinality ("width") threshold $t_{\text{abs}}$:

$$\mathcal{D}_{\text{uncertain}} = \{\boldsymbol{x} \in \mathcal{D}_1 : |C_r(\boldsymbol{x}; \hat{\tau})\| \geq t_{\text{abs}}\}.$$

We reweight the OvA surrogate loss by prediction set width $\|C_r(\boldsymbol{x}; \hat{\tau})|$ and perform a fine-tuning to explicitly emphasize these uncertain inputs:

$$\tilde{\psi}_{\text{Re-OvA}}(g_1, \ldots, g_{K+1}; \boldsymbol{x}, y, m) \;=\; \phi[g_y(\boldsymbol{x})] - \frac{\phi[-g_{K+1}(\boldsymbol{x})]}{|C_{human}(\boldsymbol{x}; \hat{\tau}_{human})|}$$

followed by re-running the dual CP to compute updated thresholds $\hat{\tau}'$ and $\hat{\tau}'_{\text{human}}$ on $\mathcal{D}_2$. Resulting coverage metrics are reported in Table 12. Finetuning on uncertain instances can raise system accuracy relative to dual CP without finetuning, but it does *not* surpass the original L2D baseline in our runs: suggesting that the OvA parameterization may capture rejector uncertainty and that post-hoc CP calibration remains a strong, distribution-free control for deferral decisions.

Table 7: Performance under uncertainty rejection across severity levels on CIFAR-10, reporting standard accuracy ($\tilde{acc}$), uncertainty rejector accuracy ($\widetilde{acc}_{m=y}$), and uncertainty classifier accuracy ($\widetilde{acc}_{\hat{y}=y}$).

| Severity | Method | $\tilde{acc}$ (%) | $\widetilde{acc}_{m=y}$ (%) | $\widetilde{acc}_{\hat{y}=y}$ (%) |
|---|---|---|---|---|
| 0 | Original | 85.02 | 86.55 | 85.02 |
| | Abstention | 86.56 | 87.84 | 71.74 |
| | Consensus | 86.63 | 87.83 | 85.15 |
| | Human Preferred | 84.27 | 86.77 | 82.68 |
| 1 | Original | 85.25 | 86.23 | 85.25 |
| | Abstention | 86.72 | 86.91 | 71.27 |
| | Consensus | 86.78 | 86.92 | 86.60 |
| | Human Preferred | 84.95 | 86.27 | 83.93 |
| | Abstention + Shifted $D_{cal}$ | 89.63 | 89.63 | 72.86 |
| | Consensus + Shifted $D_{cal}$ | 89.77 | 89.61 | 89.99 |
| | Human Preferred + Shifted $D_{cal}$ | 83.05 | 86.21 | 83.99 |
| 2 | Original | 84.05 | 86.03 | 84.05 |
| | Abstention | 85.52 | 87.02 | 71.53 |
| | Consensus | 85.57 | 87.02 | 83.79 |
| | Human Preferred | 83.60 | 86.07 | 81.44 |
| | Abstention + Shifted $D_{cal}$ | 87.84 | 88.89 | 73.11 |
| | Consensus + Shifted $D_{cal}$ | 88.01 | 88.93 | 86.77 |
| | Human Preferred + Shifted $D_{cal}$ | 81.70 | 86.00 | 81.62 |
| 3 | Original | 83.35 | 86.45 | 83.35 |
| | Abstention | 84.83 | 87.22 | 69.71 |
| | Consensus | 84.91 | 87.27 | 81.76 |
| | Human Preferred | 82.75 | 86.64 | 79.26 |
| | Abstention + Shifted $D_{cal}$ | 87.54 | 89.44 | 71.32 |
| | Consensus + Shifted $D_{cal}$ | 87.53 | 89.37 | 84.84 |
| | Human Preferred + Shifted $D_{cal}$ | 81.25 | 86.51 | 79.39 |
| 4 | Original | 83.60 | 86.32 | 83.60 |
| | Abstention | 85.54 | 87.61 | 71.18 |
| | Consensus | 85.65 | 87.70 | 83.09 |
| | Human Preferred | 83.30 | 86.49 | 79.98 |
| | Abstention + Shifted $D_{cal}$ | 87.82 | 89.17 | 72.83 |
| | Consensus + Shifted $D_{cal}$ | 88.07 | 89.37 | 86.32 |
| | Human Preferred + Shifted $D_{cal}$ | 81.90 | 86.28 | 80.09 |
| 5 | Original | 79.90 | 83.29 | 79.90 |
| | Abstention | 81.83 | 84.64 | 64.58 |
| | Consensus | 81.82 | 84.59 | 78.15 |
| | Human Preferred | 79.45 | 83.62 | 75.03 |
| | Abstention + Shifted $D_{cal}$ | 86.85 | 87.71 | 67.86 |
| | Consensus + Shifted $D_{cal}$ | 86.79 | 87.40 | 85.85 |
| | Human Preferred + Shifted $D_{cal}$ | 78.20 | 83.36 | 75.62 |
| 6 | Original | 60.58 | 74.84 | 60.58 |
| | Abstention | 73.66 | 74.64 | 12.67 |
| | Consensus | 73.63 | 74.62 | 7.49 |
| | Human Preferred | 73.81 | 74.63 | 6.15 |
| | Abstention + Shifted $D_{cal}$ | 73.08 | 73.07 | 13.63 |
| | Consensus + Shifted $D_{cal}$ | 74.48 | 74.52 | 58.82 |
| | Human Preferred + Shifted $D_{cal}$ | 74.90 | 74.63 | 6.19 |

Table 8: Performance of deferral-set workflows on HAM10000 under distribution shift. We report system accuracy ($\widetilde{acc}$), uncertainty rejector accuracy ($\widetilde{acc}_{m=y}$), and uncertainty classifier accuracy ($\widetilde{acc}_{\hat{y}=y}$).

| Severity | Method | $\widetilde{acc}$ (%) | $\widetilde{acc}_{m=y}$ (%) | $\widetilde{acc}_{\hat{y}=y}$ (%) |
|---|---|---|---|---|
| 0 | Original | 80.86 | 82.81 | 80.86 |
| | Abstention | 86.00 | 88.22 | 85.55 |
| | Consensus | 84.86 | 84.98 | 84.79 |
| | Human Preferred | 78.45 | 81.25 | 80.75 |
| 1 | Original | 75.17 | 71.99 | 75.17 |
| | Abstention | 79.73 | 72.71 | 76.34 |
| | Consensus | 80.12 | 73.08 | 86.17 |
| | Human Preferred | 76.71 | 70.42 | 82.49 |
| | Abstention + Shifted $D_{cal}$ | 82.92 | 76.09 | 80.35 |
| | Consensus + Shifted $D_{cal}$ | 82.07 | 73.79 | 88.36 |
| | Human Preferred + Shifted $D_{cal}$ | 76.77 | 69.79 | 82.81 |
| 2 | Original | 72.69 | 62.68 | 72.69 |
| | Abstention | 77.99 | 67.31 | 74.09 |
| | Consensus | 78.91 | 67.78 | 81.02 |
| | Human Preferred | 75.37 | 63.64 | 77.40 |
| | Abstention + Shifted $D_{cal}$ | 78.26 | 68.12 | 73.78 |
| | Consensus + Shifted $D_{cal}$ | 80.56 | 67.50 | 82.39 |
| | Human Preferred + Shifted $D_{cal}$ | 75.70 | 62.31 | 76.92 |
| 3 | Original | 69.21 | 63.64 | 69.21 |
| | Abstention | 74.26 | 68.13 | 72.35 |
| | Consensus | 75.56 | 65.74 | 76.18 |
| | Human Preferred | 72.69 | 67.46 | 70.39 |
| | Abstention + Shifted $D_{cal}$ | 76.56 | 74.60 | 75.56 |
| | Consensus + Shifted $D_{cal}$ | 78.78 | 69.74 | 78.77 |
| | Human Preferred + Shifted $D_{cal}$ | 74.97 | 66.35 | 69.93 |
| 4 | Original | 68.61 | 68.49 | 68.61 |
| | Abstention | 72.21 | 74.44 | 70.87 |
| | Consensus | 75.66 | 71.56 | 76.19 |
| | Human Preferred | 73.29 | 68.66 | 69.63 |
| | Abstention + Shifted $D_{cal}$ | 72.62 | 74.39 | 71.23 |
| | Consensus + Shifted $D_{cal}$ | 78.57 | 65.79 | 79.93 |
| | Human Preferred + Shifted $D_{cal}$ | 73.96 | 61.97 | 70.49 |
| 5 | Original | 68.01 | 60.61 | 68.01 |
| | Abstention | 69.72 | 66.52 | 66.67 |
| | Consensus | 78.06 | 65.38 | 80.22 |
| | Human Preferred | 74.10 | 61.59 | 71.25 |
| | Abstention + Shifted $D_{cal}$ | 68.48 | 69.59 | 66.81 |
| | Consensus + Shifted $D_{cal}$ | 79.28 | 67.87 | 81.47 |
| | Human Preferred + Shifted $D_{cal}$ | 75.17 | 65.47 | 69.92 |
| 6 | Original | 70.88 | 69.59 | 70.88 |
| | Abstention | 68.99 | 68.27 | 56.74 |
| | Consensus | 79.74 | 71.68 | 89.91 |
| | Human Preferred | 75.84 | 68.35 | 83.93 |
| | Abstention + Shifted $D_{cal}$ | 67.24 | 67.11 | 55.88 |
| | Consensus + Shifted $D_{cal}$ | 80.64 | 72.30 | 90.64 |
| | Human Preferred + Shifted $D_{cal}$ | 76.17 | 68.59 | 84.71 |

Table 9: Performance of deferral-set workflows on Hate Speech under distribution shift. We report system accuracy ($\widetilde{acc}$), uncertainty rejector accuracy ($\widetilde{acc}_{m=y}$), and uncertainty classifier accuracy ($\widetilde{acc}_{\hat{y}=y}$).

| Severity | Method | $\widetilde{acc}$ (%) | $\widetilde{acc}_{m=y}$ (%) | $\widetilde{acc}_{\hat{y}=y}$ (%) |
|---|---|---|---|---|
| 0 | Original | 92.11 | 86.88 | 92.11 |
| | Abstention | 92.16 | 86.88 | 87.58 |
| | Consensus | 92.17 | 86.88 | 95.90 |
| | Human Preferred | 92.15 | 86.88 | 95.80 |
| 1 | Original | 86.28 | 90.14 | 86.28 |
| | Abstention | 86.57 | 90.15 | 77.80 |
| | Consensus | 86.71 | 90.18 | 84.35 |
| | Human Preferred | 86.48 | 90.12 | 83.60 |
| | Abstention + Shifted $D_{cal}$ | 87.97 | 90.18 | 78.82 |
| | Consensus + Shifted $D_{cal}$ | 88.41 | 90.21 | 87.15 |
| | Human Preferred + Shifted $D_{cal}$ | 87.71 | 90.14 | 83.67 |
| 2 | Original | 79.98 | 89.48 | 79.98 |
| | Abstention | 80.51 | 89.42 | 72.36 |
| | Consensus | 80.90 | 89.46 | 75.88 |
| | Human Preferred | 80.89 | 89.45 | 74.44 |
| | Abstention + Shifted $D_{cal}$ | 84.07 | 89.83 | 74.67 |
| | Consensus + Shifted $D_{cal}$ | 85.98 | 89.93 | 83.46 |
| | Human Preferred + Shifted $D_{cal}$ | 85.25 | 89.48 | 74.49 |
| 3 | Original | 76.49 | 89.71 | 76.49 |
| | Abstention | 76.91 | 89.78 | 67.26 |
| | Consensus | 77.34 | 89.79 | 71.24 |
| | Human Preferred | 77.42 | 89.75 | 70.08 |
| | Abstention + Shifted $D_{cal}$ | 81.01 | 90.38 | 68.85 |
| | Consensus + Shifted $D_{cal}$ | 84.00 | 90.56 | 80.42 |
| | Human Preferred + Shifted $D_{cal}$ | 83.35 | 89.71 | 70.12 |
| 4 | Original | 73.18 | 89.93 | 73.18 |
| | Abstention | 73.85 | 89.81 | 64.78 |
| | Consensus | 74.37 | 89.85 | 67.11 |
| | Human Preferred | 74.66 | 89.90 | 65.40 |
| | Abstention + Shifted $D_{cal}$ | 78.49 | 90.23 | 66.74 |
| | Consensus + Shifted $D_{cal}$ | 82.59 | 90.28 | 78.50 |
| | Human Preferred + Shifted $D_{cal}$ | 82.69 | 89.93 | 65.50 |
| 5 | Original | 69.61 | 90.56 | 69.61 |
| | Abstention | 70.09 | 90.49 | 59.92 |
| | Consensus | 70.66 | 90.52 | 62.24 |
| | Human Preferred | 70.92 | 90.55 | 60.91 |
| | Abstention + Shifted $D_{cal}$ | 76.75 | 90.75 | 62.50 |
| | Consensus + Shifted $D_{cal}$ | 82.23 | 90.70 | 77.87 |
| | Human Preferred + Shifted $D_{cal}$ | 82.67 | 90.56 | 60.93 |
| 6 | Original | 66.59 | 89.05 | 66.59 |
| | Abstention | 67.30 | 89.00 | 57.37 |
| | Consensus | 67.90 | 89.03 | 59.25 |
| | Human Preferred | 68.38 | 89.02 | 57.59 |
| | Abstention + Shifted $D_{cal}$ | 71.40 | 89.05 | 57.52 |
| | Consensus + Shifted $D_{cal}$ | 77.99 | 89.29 | 72.53 |
| | Human Preferred + Shifted $D_{cal}$ | 79.52 | 89.05 | 57.60 |

Table 10: Performance under uncertainty rejection across severity levels, reporting standard accuracy, uncertainty-based rejector accuracy ($\widetilde{acc}_{m=y}$), classifier accuracy ($\widetilde{acc}_{y=y}$), non-abstention rejector accuracy, and model preferred accuracy.

| Severity | Method | $\tilde{acc}$ (%) | $\tilde{acc}_{m=y}$ (%) | $\tilde{acc}_{y=y}$ (%) |
|---|---|---|---|---|
| 0 | Baseline | 83.87 | 84.44 | 32.75 |
| | Deferral Interval | 72.57 | 92.77 | 70.06 |
| | Abstention | 93.75 | 92.77 | 74.14 |
| | Model Preferred | 92.77 | 92.77 | 70.06 |
| 1 | Baseline | 83.10 | 84.64 | 35.27 |
| | Deferral Interval | 71.90 | 92.88 | 67.65 |
| | Abstention | 93.36 | 92.88 | 74.95 |
| | Model Preferred | 92.88 | 92.88 | 67.65 |
| | Deferral Interval + Shifted $D_{cal}$ | 69.75 | 90.76 | 67.62 |
| | Abstention + Shifted $D_{cal}$ | 93.18 | 90.76 | 72.83 |
| | Model Preferred + Shifted $D_{cal}$ | 90.76 | 90.76 | 67.62 |
| 2 | Baseline | 80.65 | 83.21 | 35.35 |
| | Deferral Interval | 68.80 | 91.96 | 65.88 |
| | Abstention | 91.95 | 91.96 | 73.93 |
| | Model Preferred | 91.96 | 91.96 | 65.88 |
| | Deferral Interval + Shifted $D_{cal}$ | 70.00 | 89.56 | 66.59 |
| | Abstention + Shifted $D_{cal}$ | 92.27 | 89.56 | 72.91 |
| | Model Preferred + Shifted $D_{cal}$ | 89.56 | 89.56 | 66.59 |
| 3 | Baseline | 77.20 | 83.62 | 40.53 |
| | Deferral Interval | 62.40 | 94.55 | 58.79 |
| | Abstention | 91.75 | 94.55 | 72.58 |
| | Model Preferred | 94.55 | 94.55 | 58.79 |
| | Deferral Interval + Shifted $D_{cal}$ | 63.00 | 90.09 | 59.79 |
| | Abstention + Shifted $D_{cal}$ | 92.47 | 90.09 | 70.45 |
| | Model Preferred + Shifted $D_{cal}$ | 90.09 | 90.09 | 59.79 |
| 4 | Baseline | 76.90 | 83.78 | 42.53 |
| | Deferral Interval | 63.10 | 94.32 | 59.06 |
| | Abstention | 91.75 | 94.32 | 72.75 |
| | Model Preferred | 94.32 | 94.32 | 59.06 |
| | Deferral Interval + Shifted $D_{cal}$ | 64.80 | 92.19 | 59.33 |
| | Abstention + Shifted $D_{cal}$ | 92.78 | 92.19 | 71.33 |
| | Model Preferred + Shifted $D_{cal}$ | 92.19 | 92.19 | 59.33 |
| 5 | Baseline | 73.75 | 84.05 | 46.40 |
| | Deferral Interval | 59.80 | 90.06 | 56.97 |
| | Abstention | 90.62 | 90.06 | 72.80 |
| | Model Preferred | 90.06 | 90.06 | 56.97 |
| | Deferral Interval + Shifted $D_{cal}$ | 59.85 | 90.96 | 57.03 |
| | Abstention + Shifted $D_{cal}$ | 91.83 | 90.96 | 72.03 |
| | Model Preferred + Shifted $D_{cal}$ | 90.96 | 90.96 | 57.03 |

Table 11: *Dual CP Experiment.* We report system accuracy on dual CP on the same three datasets but do not notice performance improvement.

| Dataset | Method | Sys. Acc.(%) | Fraction Deferred (%) |
|---|---|---|---|
| CIFAR-10 | Base Model | 84.74 | 53.18 |
|  | Dual CP | 74.36 | 14.04 |
| HAM10000 | Base Model | 82.10 | 19.34 |
|  | Dual CP | 80.72 | 11.18 |
| Hate Speech | Base Model | 91.44 | 33.64 |
|  | Dual CP | 91.24 | 42.82 |

Table 12: *Dual CP Finetuning.* We report system accuracy on top of the previous dual CP experiment where $t_{\mathrm{abs}} = 1$

| Dataset | Method | Sys. Acc. (%) | Fraction Deferred (%) |
|---|---|---|---|
| HAM10000 | FT Dual CP | 81.46 | 11.12 |
| Hate Speech | FT Dual CP | 91.42 | 42.05 |

