# OpenReview forum: "Learning to Defer with an Uncertain Rejector via Conformal Prediction"
_TMLR — Accepted by TMLR_

### Review · Reviewer_cVVm · 2025-11-13

**Summary Of Contributions:**

1. The paper proposes to quantify uncertainty in the rejector of Learning-to-Defer (L2D) systems using conformal prediction (CP).
2. Instead of a binary ``defer/not-defer'' output, the rejector produces a deferral set $(0), (1), (0,1)$ or an interval over the probability that the human is correct. The motivation is that rejector uncertainty should translate into safer downstream behaviors such as abstention, consensus checking, or human-preferred routing under distribution shift.
3. The novelty of the contribution is incremental tweak on split conformal prediction applied to the rejector’s scalar probability estimate.

**Additional Comments:**

NA

**Audience:**

Yes

**Audience Explanation:**

The paper touches on an emerging direction, uncertainty quantification for deferral mechanisms, and complements ongoing work in selective prediction and conformal inference. This topic is relevant to both theoretical and applied ML researchers concerned with trustworthy human-AI collaboration. But, in the current form, the paper needs to clarify claims.

**Claims And Evidence:**

No

**Claims Explanation:**

1. The framing of the paper’s contributions is misleading. The introduction motivates the problem by claiming that an uncertain rejector can help in cases of distribution shift, since the system “cannot be sure who should handle the shifted data.” This is conceptually incorrect. Uninformative or wide conformal deferral sets arise even without any distribution shift, for example, in inherently ambiguous or borderline instances, under high variance of the rejector’s estimates, due to calibration-set scarcity, or because of noise in the human simulator. These are standard and expected behaviors of conformal prediction. They are not indicators of shift, and the paper’s narrative conflates these fundamentally different causes of uncertainty.

2. The behavior of the four L2D frameworks proposed is simply heuristics without any theoretical justification. For instance, in case of model-preferred prediction for cost saving framework, the authors say 'We expect this workflow to increase the classifier’s coverage while not substantially decreasing overall system accuracy.', without any theoretical justification.

3. Model-preferred prediction for cost saving framework is presented only qualitatively. There is no model or workflow to show how to achieve it in practice.

4. Human-Preferred Prediction under Distribution Shift framework seems distractive. Since, distribution shift is assumed, no coverage guarantees presented hold. It says 'we consider cases in which intuition leads us to believe the human is a more robust predictor than the model.' If this knowledge is available, why authors use the $C_r(x_{N+1}:\hat{\tau})$ constructed assuming exchangeability?

5. The decision rule for deferral intervals in Section 3.3, para 2, seems wrong: One possibility is to simply
abstain if the interval is on both sides of 50%. Shouldn't this be opposite?

**Requested Changes:**

1. Equation 1 seems wrong. It should have indicator comparing classifier's output with the true label. (critical change)
2. In all experiments, the target coverage level seems to be set to 90%. What is the reasoning for this choice? Can you provide or discuss the results at higher target levels? (critical change)
3. The authors should clarify and discuss the multiple independent reasons why conformal prediction may output uninformative sets, rather than implying that such uncertainty primarily signals distribution shift. (critical change)
4. Can authors describe how are they measuring coverage under abstention? I couldn't find it in the paper and thinks it is essential to add. (strengthen the work)
5. Provide detailed outline, beyond current heuristics, for Model-preferred prediction for cost saving framework as well as Human-Preferred Prediction under Distribution Shift framework. (critical change)
6. Update the decision rule for deferral intervals in Section 3.3, para 2. (critical change)

---

> ### Author Response · Authors · 2025-12-05
> **Response to Requested Changes [1]**
>
> We thank reviewer cVVm for your feedback and reviewing efforts. We are pleased to discuss the contribution. We address your
> points below:
>
> > Equation 1 seems wrong.
>
> We thank the reviewer for pointing out this typo. We take notation $\mathbb{I}[m \ne \textnormal{y}]$ to represent indicator function checking if the classifier's output and true label are equal in Equation 1, and updated the description of notation in the revision.
>
> > In all experiments, the target coverage level seems to be set to 90%. What is the reasoning for this choice? Can you provide or discuss the results at higher target levels? (critical change)
>
> We thank the reviewer for raising this point. We focused on a target coverage level of $1-\alpha = 90\%$ in the main paper because this is a standard choice in the conformal prediction literature and offers a practical trade-off between validity and efficiency (e.g., Angelopoulos and Bates, 2021). To address the reviewer’s concern, we have added experiments at higher target levels $1-\alpha \in \\{95\\%, 99\\%\\}$ for the OvA parameterization on CIFAR-10, HAM100000, and HateSpeech; the corresponding empirical coverage and average deferral-set size are reported in Appendix B.6 , Table 6. As expected, empirical coverage is close to or slightly above the target confidence levels, which is consistent with the coverage guarantees of conformal prediction. Overall, increasing the target coverage leads to higher empirical coverage together with moderately larger deferral sets. Due to time constraints in the rebuttal period, we restricted these additional runs to OvA and did not repeat the full validation sweep for all parameterizations; a more exhaustive study across both OvA and A-SM at multiple coverage levels can be followed in a subsequent version of the paper.
>
> Anastasios N Angelopoulos and Stephen Bates. A gentle introduction to conformal prediction and distribution-free uncertainty quantification. arXiv preprint arXiv:2107.07511, 2021
>
> Anastasios Nikolas Angelopoulos, Stephen Bates, Michael Jordan, and Jitendra Malik. Uncertainty sets for image classifiers using conformal prediction. In International Conference on Learning Representations, 2021. URL https://openreview.net/forum?id=eNdiU_DbM9.

---

> ### Author Response · Authors · 2025-12-05
> **Response to Requested Changes [2]**
>
> > The authors should clarify and discuss the multiple independent reasons why conformal prediction may output uninformative sets, rather than implying that such uncertainty primarily signals distribution shift. (critical change)
>
>
> We thank the reviewer for their critical and insightful feedback regarding the framing of our contributions. We fully agree that conformal prediction can produce large sets even without any shift, for several independent reasons: (i) intrinsically ambiguous or borderline instances, (ii) high variance or misspecification in the underlying score function, (iii) limited calibration data, and (iv) noisy or multi-valued ground truth for the human simulator. However, we want to clarify our motivation and calibrated rejector.
> 1. Clarifying the Sources and Impact of Uncertainty: Our intention was not to claim that large deferral sets uniquely diagnose distribution shifts. We acknowledge the reviewer’s point that wide conformal sets (e.g., $C_r(x) = \\{0, 1\\}$) can arise from multiple distinct factors, including inherent data ambiguity (aleatoric uncertainty), calibration data scarcity (epistemic uncertainty), and label noise, instead of just distribution shift. We have updated the Introduction and Method sections to make this point explicit and to avoid language that conflates wide sets with shift detection. Rather, we argue that for the learning-to-defer system, all of the reviewer’s scenarios share the same operational meaning: the rejector cannot confidently determine whether the expert or the model is more likely to be correct. While the source of the uncertainty varies, the implication for the Learning-to-Defer system remains consistent: when the rejector is unsure whether the human or the model is the superior decision-maker, the system should avoid making a confident routing decision.
> In addition, for calibration-set scarcity, to demonstrate the calibration threshold $\hat{\tau}$ is well-fitted, we show the size conformal set is efficient in the in-distribution case. We empirically present calibration sets provide a good quality as producing conformal sets of average size from 1.01 to 1.37 in Section 5.1.
>
> 2. Rejector is properly calibrated: We are with the reviewer that conformal coverage guarantees rely on exchangeability, which breaks under distribution shift. However, we contend that conformal sets remain a useful heuristic for uncertainty as the underlying rejector is effectively calibrated. As established in prior work (e.g., Ovadia et al., 2019), well-calibrated models tend to exhibit increased entropy (lower confidence) as inputs drift from the training distribution. Specifically for our L2D:
> * As shift increases, the rejector tends to harder to distingish between 0 and 1, whose estimated probability of expert correctness, $\hat{P}(m=y|x)$, tends to converge toward 0.5 (maximum entropy at extreme cases).
> * Consequently, the conformal set construction, which relies on these scores, naturally widens to $\\{0, 1\\}$ to maintain coverage.
> To empirically substantiate this claim, we report the entropy of the rejector's correctness probability $\hat{P}(m=y|x)$.
>
> These results demonstrate that while the conformal coverage guarantee degrades under shift, the underlying rejector successfully detects the increased uncertainty. Consequently, the widening of the conformal deferral set is not an arbitrary failure, but a calibrated response to the increasing difficulty of the test samples. This validates our proposed workflow.
>
> | Dataset | Severity 0 | Severity 1 | Severity 2 | Severity 3 | Severity 4 | Severity 5 | Severity 6 |
> | ----- | ----- | ----- | ----- | ----- | ----- | ----- | ----- |
> | **CIFAR-10** | 0.3122 | 0.3394 | 0.3691 | 0.3988 | 0.4185 | 0.4227 | -- |
> | **HAM10000** | 0.1280 | 0.1063 | 0.1374 | 0.1457 | 0.1944 | 0.2645 | 0.2736 |
> | **HateSpeech** | 0.3078 | 0.3487 | 0.3650 | 0.3714 | 0.3776 | 0.3834 | 0.3896 |
>
>
> Yaniv Ovadia, Emily Fertig, Jie Ren, Zachary Nado, Sebastian Nowozin, Joshua Dillon, Balaji Lakshminarayanan, and Jasper Snoek. Can you trust your model’s uncertainty? evaluating predictive uncertainty under dataset shift. Advances in Neural Information Processing Systems, 32, 2019.
>
> > Can authors describe how are they measuring coverage under abstention? I couldn't find it in the paper and thinks it is essential to add. (strengthen the work)
>
> I assume you are talking about coverage related to the ratio of test points deferred. We measure by the fraction of points for which the system does not abstain. If you refer to the coverage related to CP’s guarantee. Conformal coverage is measured by the ratio of correct instances predicted by the human or model over the instances which the system does not abstain.

---

> ### Author Response · Authors · 2025-12-05
> **Response to Requested Changes [3]**
>
> > 5. Provide detailed outline, beyond current heuristics, for Model-preferred prediction for cost saving framework as well as Human-Preferred Prediction under Distribution Shift framework. (critical change)
>
> We thank the reviewer for this critical insight. We agree that describing these workflows merely as heuristics. Our core theoretical contribution is at the level of uncertainty quantification for the rejector: we provide conformal prediction guarantees for the binary variable $I[m=y]$ (expert correctness), yielding calibrated deferral sets and intervals. The four workflows are decision rules defined on top of these uncertainty objects, and we intend them as practically-motivated triage policies whose performance is evaluated empirically, rather than as theorems about optimality.
> In the revision, we have visualized both in Figure 2 and formalized these workflows as decision-theoretic rules that minimize specific loss functions (cost-sensitive loss and robust risk, respectively) in Appendix A2.
>
> > 6. Update the decision rule for deferral intervals in Section 3.3, para 2. (critical change)
>
> We thank the reviewer for pointing out this typo and change to clarify. We expect to abstain when the interval across 0.5 which means that the deferral is in the middle area and uncertain to determine 0 or 1. We updated the equation to $0.5 \in [b_{l}(\mathbf{x}), b_{r}(\mathbf{x}))$. This means $ [b_{l}(\mathbf{x}) <=  0.5 <= b_{r}(\mathbf{x}))$.

---

### Review · Reviewer_QR8o · 2025-11-20

**Summary Of Contributions:**

I think the main contribution of this paper is introducing conformal prediction into the rejector of L2D and constructing deferral sets with statistical coverage guarantees.

The paper estimates the probability of expert correctness, applying conformal prediction yields deferral sets {0}, {1}, {0, 1}, this leads to novel three alternative L2D workflows: abstention, consensus prediction, and human-preferred routing.

The theoretical innovation is limited, but the work fills a gap in uncertainty quantification within the L2D studies and identifies a real limitation: the deferral mechanism can fail silently under distribution shift. Introducing distribution-free UQ to this component is sensible and fills a gap in the literature. Moreover, the workflows make use of uncertainty in routing, not classification. I think it conceptually expands the L2D design space.

I think it is a shame that the CP target is only expert correctness, not the Bayes-optimal deferral rule, the proposed CP construction only calibrates the expert side.

**Additional Comments:**

N/A

**Audience:**

Yes

**Audience Explanation:**

I previously read a paper titled “CP-Router: An Uncertainty-Aware Router Between LLM and LRM,” and I feel it follows a somewhat similar paradigm. I think, especially for researchers studying decision routing, deferral systems, or uncertainty in human-AI teams, there is clear audience interest.

I believe that UQ does not have to be applied only to predicting the answer itself, it can also be applied to predicting who should provide the answer. This is precisely what the paper addresses, and introducing uncertainty into the system’s decision-routing layer can indeed be combined with risk control to improve the safety of the entire pipeline.

The work demonstrates that the rejector itself requires credibility assessment. By using conformal prediction to generate deferral sets or uncertainty intervals, the system can explicitly express I don’t know whom to assign this instance to, which makes abstention, consensus prediction, or human-preferred strategies natural and justified.

In other words, it proposes a new safety paradigm: when the system is uncertain about which source is more trustworthy, it can defer the decision rather than proceeding blindly.

**Claims And Evidence:**

No

**Claims Explanation:**

The core claims about achieving nominal coverage, constructing compact deferral sets, and improving robustnessvia abstention and consensus workflows are backed by clear experiments across three datasets, with appropriate reporting of coverage, set size, accuracy, and deferral rates. Meanwhile, the OOD experiments systematically vary shift severity and include reasonable baselines, this demonstrates that conformalized rejectors can yield more stable performance under moderate shift.

However, as to safety or being closer to the Bayes-optimal rejector, I think the method calibrates expert correctness, not the full deferral decision, and there is no theoretical guarantee under distribution shift, there are only empirical results.

**Requested Changes:**

1. Please clarify "an indicator function that checks if the prediction and label are equal" in Eq.(1).

2. I think what truly needs calibration is not expert correctness alone, but the comparison between expert correctness and model correctness. Instead, the paper applies CP to expert correctness as a single variable. As a result, the coverage guarantee obtained by the method does not correspond to the optimal deferral decision in any meaningful sense. In other words, the paper addresses a proxy for the rejector but not the rejector itself.

3. The deferral set tends to become wider under distribution shift, showing some degree of shift-awareness, but this behavior is entirely empirical. While the experiments show seemingly reasonable behavior, the paper offers no explanation for why the rejector score becomes less confident under shift, nor whether certain data properties or model parameters influence this phenomenon. Just OOD robustness is observed but not understood.

---

> ### Author Response · Authors · 2025-12-05
> **Response to comment and Requested Changes [1]**
>
> Thank you, QR8o, for taking the time to review our work. We are pleased that you find the paper fills an important gap, and we appreciate you pointing us to CP-Router; in the revision, we have also added a paragraph discussing related LLM-based work.
>
> > Please clarify "an indicator function that checks if the prediction and label are equal" in Eq.(1).
>
> Here, $\mathbb{I}\[m \neq y \]$ denotes the indicator function, which equals 1 if the human prediction does not match the true label and 0 otherwise. We updated its description for more detail in the revision.

---

> ### Author Response · Authors · 2025-12-05
> **Response to Requested Changes [2]**
>
> >  the paper addresses a proxy for the rejector but not the rejector itself.
>
> We thank the reviewer for the thoughtful distinction between calibrating expert correctness and calibrating the full deferral decision. We agree that our method formally provides guarantees on expert correctness $\mathbb{P} ( \mathbb{I} [ m\_{N+1}= y\_{N+1} ]  \in  C_{r}(\mathbf{x}_{N+1}; \uptau ) ) $  rather than the Bayes-optimal deferral decision itself $\mathbb{P}(r^{*}(\mathbf{x}\_{N+1} ) \in C\_{r} (\mathbf{x}\_{N+1}; \uptau ) )$. We address this distinction in our "Ideal vs. Practical Construction" section, i.e. Section 3. We argue that focusing on expert correctness is the necessarily rigorous and practical path forward for three reasons: theoretical constraints on comparing probabilities, alignment with established L2D literature, and empirical evidence from our Dual CP experiments.
>
> (1) Theoretical Constraints on the "Ideal" Construction: As noted in Section 3, the Bayes-optimal rejector relies on comparing the ground-truth conditional probabilities of the expert and the classifier $r^*(x) = \mathbb{I}\[ \mathbb{P}(m=y|x) \ge \max_{y} \mathbb{P}(y|x) \]$. Constructing a prediction set or interval that guarantees this comparison requires high-fidelity estimates of two specific conditional probabilities for a single instance. Obtaining such estimates for one-off events is known to be impossible without strong distributional assumptions (Roth et al., 2023, Johnson et al., 2024). Consequently, we focus on the "Practical Construction" of calibrating expert correctness, which allows for valid, distribution-free finite-sample guarantees.
>
> (2) Alignment with Established Definitions of Calibration in L2D: Our approach adheres to the standard definition of calibration established in the L2D literature. Most notably, Verma & Nalisnick (2022) and Cao et al. (2024) define and demonstrate L2D calibration specifically as the soundness of the expert correctness probability. They formally define calibration (Eq. 7 in Verma & Nalisnick’s work ) as: $\mathbb{P}(m=y \mid p_m(x)=c) = c$
> OvA work observed that previous Softmax parameterizations (Mozannar & Sontag, 2020) produce unbounded estimators and unuseful proxies. Crucially, they focus entirely on deriving a valid, calibrated estimator for $\mathbb{P}(m=y|x)$, i.e. OvA and explicitly show that  $\mathbb{P}(m=y|x)$ is objective to represent the expert's degree of superiority to the classifier for safe deferral. By applying conformal prediction to this exact variable ($\mathbb{I}\[m=y\]$), our work provides finite-sample guarantees on the very quantity the L2D literature has identified as the primary source of miscalibration, consistent with the field's precedent.
>
> (3) Empirical Comparison with Dual Calibration To address the reviewer's intuition, we explored a "Dual CP" approach (now in Appendix D.1), where we constructed separate prediction sets for both the classifier and the expert to approximate the "full" deferral decision. However, we found that this approach did not yield empirical improvements over our proposed method of calibrating expert correctness (Table 11 now). We hypothesize this is because the "full" decision requires intersecting two uncertainty sets, leading to overly conservative behavior (loss of efficiency) without gaining decision accuracy. This reinforces that calibrating expert correctness is not only theoretically more tractable but also empirically sufficient for robust performance.
> We have updated the manuscript to clarify these points and explicitly reference the precedent set by previous calibration work in L2D. As a result, we treat the ideal construction as a conceptual target and leave it as an open problem.
>
> Rajeev Verma and Eric Nalisnick. Calibrated Learning to Defer with One-vs-All Classifiers. In Proceedings of the 39th International Conference on Machine Learning, 2022.
>
> Yuzhou Cao, Hussein Mozannar, Lei Feng, Hongxin Wei, and Bo An. In defense of softmax parametrization for calibrated and consistent learning to defer. Advances in Neural Information Processing Systems, 36, 2024.
>
> Aaron Roth, Alexander Tolbert, and Scott Weinstein. Reconciling individual probability forecasts. In Proceedings of the 2023 ACM Conference on Fairness, Accountability, and Transparency, pp. 101–110, 2023.
>
> Daniel D Johnson, Daniel Tarlow, David Duvenaud, and Chris J Maddison. Experts don’t cheat: learning what you don’t know by predicting pairs. In Proceedings of the 41st International Conference on Machine Learning, pp. 22406–22464, 2024

---

> ### Author Response · Authors · 2025-12-05
> **Response to Requested Changes [3]**
>
> > Just OOD robustness is observed but not understood.
>
> We thank the reviewer for highlighting the need to explain. We agree that our original discussion of the widening deferral sets under distribution shift was mostly empirical, and we are happy to clarify the mechanism. We have clarified in the revised manuscript why the conformal deferral sets widen under shift by adding Appendix A1.
>
> The phenomenon is a behavior of how Split-CP interacts with model degradation under shift. Recall that in Section 3.1 we apply split conformal prediction to the binary meta-label $I[m=y] \in \\{0,1\\} $. By Eq. (8), the conformal deferral set is
> $$
> C_r(x;\hat\tau)=
> \begin{cases}
> \\{0\\}, & \hat p(m=y\mid x)\le \hat\tau,\\\\
> \\{1\\}, & \hat p(m=y\mid x)\ge 1-\hat\tau, \\\\
> \\{0,1\\}, & \text{otherwise},
> \end{cases}
> $$
>
> Our CP produces the uncertain set $\\{0, 1\\}$ if and only if the rejector's score $\hat{p}(m=y|x)$ falls within the interval $\[\hat{\tau}, 1-\hat{\tau}\]$, where $1-\hat{\tau}$ is the calibrated threshold, $\hat{\tau} < \hat{p}(m=y|x) < 1-\hat{\tau}$.
> On in-distribution (ID) data, the rejector is generally confident, meaning $\hat{p}(m=y|x)$ concentrates near 0 or 1, falling outside this "ambiguity region", the interval $[\hat{\tau}, 1-\hat{\tau}]$.
> Under distribution shift, the i.i.d. assumption required for coverage guarantees is violated and degrades the rejector’s predictive performance. As the test data diverges from the training distribution, the source-trained rejector 's epistemic tends to produce less distinguishable probabilities on shifted inputs. As $\hat{p}(m=y|x)$ shifts from the extrema toward $0.5$, a larger mass of samples falls into the interval $\[\hat{\tau}, 1-\hat{\tau}\]$. This mechanically forces the CP procedure to output the wider set $\\{0, 1\\}$, effectively translating OOD degradation into explicit deferral uncertainty.

---

### Review · Reviewer_gQn7 · 2025-11-21

**Summary Of Contributions:**

The paper investigates uncertainty estimation within the Learning-to-Defer (L2D) paradigm, which allocates prediction tasks between human and machine decision-makers. The authors highlight that the performance of L2D critically depends on the quality of the rejector function, which may be misspecified or poorly fitted in practice. To address this, they propose applying conformal prediction to the rejector, allowing it to produce prediction sets or intervals with user-specified confidence levels, rather than a simple binary defer decision. Experimental results on diverse tasks, including image classification and hate speech detection, suggest that quantifying rejector uncertainty can lead to safer decision-making through two forms of selective prediction.

The strengths are that they deal with an important topic and demonstrate various applications. The weaknesses are that the paper does not explain certain points well enough or go into sufficient detail, and the experiments reveal weaknesses.

**Additional Comments:**

- I think the background section is good in terms of length and provides explanations of the basics at an appropriate level.
- I think the various approaches to covariate shift are well chosen.

**Audience:**

Yes

**Audience Explanation:**

Yes, I think other researchers are interested in this work because it is an interesting topic and, as it seems, the authors present different methodological approaches. However, I find the experimental design to be still too incomplete and insufficiently discussed.

**Claims And Evidence:**

No

**Claims Explanation:**

- Human-preferred prediction under distribution shift is mentioned as a method for making decisions, but how do I even know when I have distribution shifts, or is it just an example of a more passive approach where humans are more likely to be consulted?
- Uncertainty information could also be retrieved directly from the classifier; the authors use CP as a meta model which already uses CP as uncertainty quantifier. As a final step, they consider an uncertainty interval for the rejector, i.e., uncertainty from the meta model. I don't see the reasoning behind this step.
Moreover, the authors state: "We expect this workflow to increase the classifier’s coverage while not substantially decreasing overall system accuracy." The experiments from section 5.4 are in the appendix, and I don't see any coverage rates.
- The Conformal Prediction for L2D in the related work section is described very briefly, and comparisons are only made with Liu et al. 2022 in the experiments. Why could no comparisons be made with the other baselines, or are they missing from the experiments?
- For each of the three use cases, one dataset is utilized. I think it is beneficial that there are different applications, but for the generalization of the method, two datasets per application would be helpful.
- When coverage decreases, accuracy improves. This suggests that the authors' method works, but at the same time, it also happens as soon as I do not classify these samples due to some uncertainty function. In order to demonstrate the improvements of the method compared to other baselines, I am missing other evaluations that show that higher accuracy can be achieved with greater coverage.
- I find the discussion lacking in the numerical results, i.e., a result is usually explained in 1-2 sentences, but there is no further discussion of what it means.

**Requested Changes:**

critical changes:
- The abstract is very concise and difficult to understand if you are not researching this topic.
- The authors write: "See the supplementary materials for additional details about more algorithm detail, training hyperparameters and backbone architectures" I couldn't find the information.
- The authors express their improvements in percentages, but these are percentage points.
- I am missing a decisive baseline, namely to use the uncertainty of the classifier directly (i.e., without a rejector meta model) and to set a threshold on the sigmoid function in the range where no decision is made.
- I find it really difficult to compare accuracy vs. ratio deferred, as accuracy often increases when fewer decisions are made. Perhaps a plot directly comparing the two would be useful for identifying differences in the methods.
- Table 3 compares different methods than those in the main paper, which I find somewhat confusing.
- The authors write: "Resulting coverage and deferral-rate metrics are reported in Table 5." I see only accuracy.


minor changes:
- At the beginning of the introduction, it would be helpful to mention specific examples of L2D applications.
- In figures and tables, sometimes very large font size, then very small again - should be consistent and match the text size.
- Mean and standard deviation are given, but not how many splits (cross validation) were used to obtain the results.
- Sometimes it is written as “system accuracy,” then only “accuracy,” or “standard accuracy,” which all describe the same thing (as I understand it), so it should be kept consistent.
- I find it unusual that Fig. 2 is divided by text and across pages - somewhat confusing; I would prefer it as single figures.
- Why were no experiments performed for Cifar10 in Table 5?

---

> ### Author Response · Authors · 2025-12-05
> **Response for comments**
>
> We appreciate the reviewer gQn7’s comments and the opportunity to clarify the role of conformal prediction (CP) in our framework. We are glad to hear you found the background section and covariate shift are good.
>
> > How do I even know when I have distribution shifts, or is it just an example of a more passive approach where humans are more likely to be consulted?
>
> Human-preferred prediction under distribution shift is an example of a more passive approach where we attempt to recapture full coverage.
>
> > 2(a) Reasoning to not use the classifier’s uncertainty instead of a “meta” CP on the rejector?
>
> Our goal is to quantify uncertainty in the rejector, i.e., in the event that the expert is correct, rather than in the classifier’s prediction. We add the selective prediction baseline to answer this empirically, where both rejector trained with OvA and A-SM and our proposed workflow outperform the selective prediction baseline.
>
> > 2 (b) Reasoning to introduce deferral intervals in addition to deferral sets?
>
> The conformal deferral set $C_r(x;\hat{\tau}) \in \{\{0\}, \{1\}, \{0,1\}\}$ is a discrete, tri-valued summary of rejector uncertainty (``use the model / use the expert / uncertain''), which suffices for the abstention and consensus workflows in Section~3.2. However, the underlying quantity of interest is the continuous probability $\theta(x) = P(m = y \mid x) \in [0,1]$.
> Following Barber (2020), we reinterpret the problem as binary regression on $\mathbb{I}[m = y]$ and construct a conformal interval $[b_\ell(x), b_r(x)]$. This interval is strictly richer than the deferral set: it supports robust, Bayes-inspired routing rules that compare a calibrated interval for expert accuracy against the classifier's confidence. For instance, we can defer only when the lower endpoint $b_\ell(x)$  exceeds $\max_{y} P(y \mid x)$, ensuring that even in the worst case within the conformal interval the expert is more likely to be correct than the model. Conceptually, deferral intervals therefore enable cost-aware and safety-critical triage policies that are not expressible using only the discrete deferral set. However, as we discussed in the main paper, the interval is too large to be meaningful.
>
>  "We expect this workflow to increase the classifier’s coverage while not substantially decreasing overall system accuracy." We refer to the workflow of Model-Preferred Prediction. In this workflow, instances would be directed to model when to abstain. Model preferred workflow would lead to a 100% coverage, We updated coverage in Table 5 (now Table 12).
>
> Rina Foygel Barber. Is distribution-free inference possible for binary regression? Electronic Journal of Statistics, 14(2):3487 – 3524, 2020. doi: 10.1214/20-EJS1749. URL https://doi.org/10.1214/20-EJS1749.
>
> > 3 Why could no comparisons be made with the other baselines, or are they missing from the experiments?
>
> We thank the reviewer for suggesting a decisive baseline. Our experiments compared against several baselines: OvA and A-SM as non-conformal L2D baselines, and Finetuned on Shift and Ensemble as uncertainty and robustness baselines. In the revision, following this suggestion, we additionally introduce a Selective Prediction baseline (thresholding the classifier's softmax) and describe it in more detail in our response to the requested changes and in the updated experimental section.
>
> > I find the discussion lacking in the numerical results, i.e., a result is usually explained in 1-2 sentences, but there is no further discussion of what it means.
>
> We thank you for your suggestion. We added discussions to both Section 5.2 and 5.3.

---

> ### Author Response · Authors · 2025-12-05
> **Response for requested changes [1]**
>
> critical changes:
> > The abstract is very concise and difficult to understand if you are not researching this topic.
>
> We updated the abstract to include more detail about the topic of conformal prediction and learning to defer.
>
> > The authors write: "See the supplementary materials for additional details about more algorithm detail, training hyperparameters and backbone architectures" I couldn't find the information.
>
> We thank the reviewer for catching this. We have now added a dedicated section, Appendix B1-4, which lists datasets, model configurations, and covariate shift simulation details.
>
> > The authors express their improvements in percentages, but these are percentage points.
>
> We sincerely thank the reviewer for their keen eye. We agree that using "percentage points" is the correct and precise terminology, and we have updated Section 5.2 to strictly adhere to this terminology.
>
> > I am missing a decisive baseline, namely to use the uncertainty of the classifier directly (i.e., without a rejector meta model) and to set a threshold on the sigmoid function in the range where no decision is made.
>
> We thank the reviewer for suggesting this decisive baseline. To address this, we implemented Selective Prediction following the protocol defined in Mozannar et al., 2023 and Geifman & El-
> Yaniv, 2017. This baseline utilizes the classifier's confidence (maximum softmax probability) directly, without a separate rejector meta-model. We determine the optimal threshold $\tau$ on the held-out calibration set to maximize total system accuracy.
>
> We have updated Figure 3 (Figure 2 in original manuscript) and Figure 4 in Appendix B1 to include the selective prediction baseline. While Selective Prediction improves upon the classifier-alone baseline, our proposed Conformal L2D method generally outperforms it. This suggests that relying solely on classifier uncertainty is insufficient; explicitly modeling the expert's correctness (as done in L2D) allows for more complementary deferral decisions than confidence thresholding alone.
> We observe that while Selective Prediction generally improves upon the base model by filtering low-confidence predictions, it underperforms compared to the Conformal L2D approaches (OvA and A-SM).
>
> Hussein Mozannar, Hunter Lang, Dennis Wei, Prasanna Sattigeri, Subhro Das, and David Sontag. Who Should Predict? Exact Algorithms For Learning to Defer to Humans. In Proceedings of the 26th International Conference on Artificial Intelligence and Statistics, 2023
>
> Yonatan Geifman and Ran El-Yaniv. Selective classification for deep neural networks. In Advances in Neural Information Processing Systems, volume 30, 2017. URL https://proceedings.neurips.cc/
> paper_files/paper/2017/file/4a8423d5e91fda00bb7e46540e2b0cf1-Paper.pdf.
>
> > I find it really difficult to compare accuracy vs. ratio deferred, as accuracy often increases when fewer decisions are made. Perhaps a plot directly comparing the two would be useful for identifying differences in the methods.
>
> We appreciate reviewer gQn7’s feedback and assume this is talking about Figure 2 in the initial manuscript. In response, we provide additional plots in Figure 4 in the same plotting style for this comparsion.
>
> Some figures exclude Ensembles baseline as their plots are distant from major methods. For instance, in HAM10000, the deferral ratio of all ensembles remains 0.
>
> > Table 3 compares different methods than those in the main paper, which I find somewhat confusing.
>
> We thank the reviewer for pointing out this source of confusion. The new metrics of uncertainty-based rejector accuracy and uncertainty classifier accuracy dive deeper into how abstention and consensus identify uncertain inputs and abstain from prediction. Table 3 evaluates the interval-based variant introduced in Section 3.3 together with system accuracy. While the main paper already demonstrates this capability of the deferral set by showing improvements in overall system accuracy. In the revision, the new appendix tables, i.e Table 7-9, explicitly present the performance of decision making with deferral sets across severity levels for the same methods as in the main paper.
>
> > The authors write: "Resulting coverage and deferral-rate metrics are reported in Table 5." I see only accuracy.
>
> We thank the reviewer for pointing this out. We updated Table 5 now Table 12.

---

> ### Author Response · Authors · 2025-12-05
> **Response for requested changes [2]**
>
> minor changes:
> > At the beginning of the introduction, it would be helpful to mention specific examples of L2D applications.
>
> We updated the introduction section. Below is an example we add:
> As a concrete example, consider automated diagnosis of pigmented skin lesions. For a given dermatoscopic image, our L2D system either outputs a prediction together with a user-specified conformal confidence guarantee, or abstains when the current image features alone are insufficient to meet this guarantee. In the latter case, a clinician could request further information such as running additional lab tests or biopsies, before making the final decision. This thereby makes more trustworthy and clinically safer decisions.
>
> > In figures and tables, sometimes very large font size, then very small again - should be consistent and match the text size.
>
> We thank reviwer gQn7's feedback. We tried to manage the table and figure font size in the revision manuscript.
>
> > Mean and standard deviation are given, but not how many splits (cross validation) were used to obtain the results.
>
> We had 5 splits cross validation of the experiment.
>
> > Sometimes it is written as “system accuracy,” then only “accuracy,” or “standard accuracy,” which all describe the same thing (as I understand it), so it should be kept consistent.
>
> We thank the reviewer for catching this inconsistency. Yes, we refer them to the same thing. In the revision, we now use the term “system accuracy” consistently throughout the manuscript. We updated in the new revision.
>
> > I find it unusual that Fig. 2 is divided by text and across pages - somewhat confusing; I would prefer it as single figures.
>
> We appreciate the reviewer’s suggestion and have revised the layout so that all panels of Fig. 2 (now Fig 3) now appear together on a one and followed page.
>
> > Why were no experiments performed for Cifar10 in Table 5?
>
> We found empirically hard and uninterestingly to simulate the human expert annotator in the setting of CIFAR10.

---

### Decision · Action_Editor_imWx · 2026-01-05

**Recommendation:** Accept as is

**Audience:**

Yes

**Audience Explanation:**

After the discussion phase, all three reviewers agree that, while the paper contains weaknesses, its subject matter and empirical results are of interest to the TMLR community.

**Claims And Evidence:**

Yes

**Claims Explanation:**

Two of the three reviewers agree that, while incremental, the paper provides sufficient empirical evidence for most of its claims, especially given the changes during the discussion period.
I agree with them.